# Autoinducer-2 and bile salts induce c-di-GMP synthesis to repress the T3SS via a T3SS chaperone

Shuyu Li[1,3], Hengxi Sun[1,3], Jianghan Li[1], Yujiao Zhao[1], Ruiying Wang[1], Lei Xu [1], Chongyi Duan[1], Jialin Li[1], Zhuo Wang[1], Qinmeng Liu[1], Yao Wang [1], Songying Ouyang [2] ✉, Xihui Shen [1] ✉ & Lei Zhang [1] ✉

Cyclic di-GMP (c-di-GMP) transduces extracellular stimuli into intracellular responses, coordinating a plethora of important biological processes. Low levels of c-di-GMP are often associated with highly virulent behavior that depends on the type III secretion system (T3SS) effectors encoded, whereas elevated levels of c-di-GMP lead to the repression of T3SSs. However, extracellular signals that modulate c-di-GMP metabolism to control T3SSs and c-di-GMP effectors that relay environmental stimuli to changes in T3SS activity remain largely obscure. Here, we show that the quorum sensing signal autoinducer-2 (AI-2) induces c-di-GMP synthesis via a GAPES1 domain-containing diguanylate cyclase (DGC) YeaJ to repress T3SS-1 gene expression in *Salmonella enterica* serovar Typhimurium. YeaJ homologs capable of sensing AI-2 are present in many other species belonging to *Enterobacterales*. We also reveal that taurocholate and taurodeoxycholate bind to the sensory domain of the DGC YedQ to induce intracellular accumulation of c-di-GMP, thus repressing the expression of T3SS-1 genes. Further, we find that c-di-GMP negatively controls the function of T3SSs through binding to the widely conserved CesD/SycD/LcrH family of T3SS chaperones. Our results support a model in which bacteria sense changes in population density and host-derived cues to regulate c-di-GMP synthesis, thereby modulating the activity of T3SSs via a c-di-GMP-responsive T3SS chaperone.

Cyclic-di-GMP (c-di-GMP) is an important second messenger that is present in the majority of bacteria and controls a wide range of biological processes, including (but not limited to) motility, biofilm formation, virulence, cellular differentiation, and stress adaptation[1–4]. C-di-GMP is enzymatically synthesized by diguanylate cyclases (DGCs) containing a GGDEF domain and degraded by c-di-GMP-specific phosphodiesterases (PDEs) harboring an EAL or HD-GYP domain[1].

While many bacterial species encode dozens of DGCs and PDEs within their genomes, many of these proteins contain periplasmic or cytoplasmic sensory domains linked to the enzymatic domains[5–8]. Cellular levels of c-di-GMP are modulated by changes in the expression or activity of c-di-GMP-metabolizing enzymes (CMEs) in response to quorum sensing (QS) signals, environmental stimuli, or host-derived cues[4–12].

[1]State Key Laboratory of Crop Stress Biology for Arid Areas, Shaanxi Key Laboratory of Agricultural and Environmental Microbiology, College of Life Sciences, Northwest A&F University, Yangling, Shaanxi 712100, China. [2]The Key Laboratory of Innate Immune Biology of Fujian Province, Provincial University Key Laboratory of Cellular Stress Response and Metabolic Regulation, Biomedical Research Center of South China, Key Laboratory of OptoElectronic Science and Technology for Medicine of Ministry of Education, College of Life Sciences, Fujian Normal University, Fuzhou, China. [3]These authors contributed equally: Shuyu Li, Hengxi Sun. ✉e-mail: ouyangsy@fjnu.edu.cn; xihuishen@nwsuaf.edu.cn; zhanglei0075@nwsuaf.edu.cn

QS is a cell-cell communication mechanism dependent on cell density that coordinates bacterial group behaviors such as bioluminescence, biofilm formation, and virulence factor production[7,13]. QS depends on the production, release, and detection of signaling molecules called autoinducers[4,8]. Whereas most known autoinducers are species-specific, autoinducer-2 (AI-2) is reported to be a universal QS signal that can be produced by both Gram-negative and Gram-positive bacteria and used for both interspecies and intraspecies communication[7]. QS signals, including acyl-homoserine lactone, AI-2, and diffusible signal factor (DSF), can be integrated into the c-di-GMP signaling network by directly or indirectly regulating the expression or activity of CMEs[4,7,8,12]. A few specific environmental signals that influence bacterial c-di-GMP metabolism have also been identified, including oxygen, nitric oxide, light, temperature, arginine, and norspermidine[4–6]. Moreover, physiologically relevant host cues such as bile and bicarbonate have been shown to regulate c-di-GMP levels in *Vibrio cholerae*[10,14]. Nevertheless, extracellular signals that can directly modulate the activity of CMEs and the underlying mechanisms remain largely undefined.

While intracellular c-di-GMP levels are controlled in time and space, the influence of this ubiquitous signaling molecule on bacterial physiology and behavior relies on its sensors, including mRNA riboswitches[15] and target proteins such as transcriptional regulators[1,16], enzymes[17,18], metabolic switch[19] and adapter proteins[20,21]. C-di-GMP signaling has been widely reported to regulate virulence in various bacterial pathogens of mammalian hosts[22–25]. Generally, low levels of c-di-GMP are associated with a motile and highly virulent state, whereas elevated levels of c-di-GMP favor biofilm formation and chronic infections[1,22,23]. Additionally, recent studies have shown that c-di-GMP negatively controls the type III secretion systems (T3SSs) of various pathogenic bacteria[22–24]. For example, in *S.* Typhimurium, high levels of c-di-GMP lead to reduced secretion of T3SS-1 effectors that contribute to the invasion of epithelial cells, including SipA, SipB, SopB, and SopE2[23,24]. However, c-di-GMP sensors through which this second messenger exerts its effect on T3SSs remain unidentified.

Here, we show that the DGC YeaJ senses AI-2 via its GAmmaproteobacterial PEriplasmic Sensor (GAPES1) domain to increase intracellular c-di-GMP levels, thus leading to repression of the T3SS-1. We further find that bile components taurocholate and taurodeoxycholate repress T3SS-1 gene expression by modulating the intracellular concentration of c-di-GMP via the DGC YedQ. Moreover, we report the discovery of the CesD/SycD/LcrH family of T3SS chaperones as c-di-GMP effectors in a variety of bacterial pathogens within the phylum *Proteobacteria*, thus revealing a widely conserved mechanism through which c-di-GMP signal input modulates the activity of T3SSs. The present study significantly extends our knowledge of the mechanisms involved in c-di-GMP metabolism and recognition in pathogenic bacteria.

## Results

### AI-2 induces c-di-GMP synthesis by acting as an activator of YeaJ in *S.* Typhimurium

Although AI-2 is the well-known QS molecule produced by enteric bacteria, including *E. coli* and *S.* Typhimurium[7,26], its physiological role in these bacteria remains poorly understood. Consistent with several previous studies[27,28], we found that deletion of *luxS* led to significantly reduced biofilm formation in *S.* Typhimurium (Supplementary Fig. 1a). However, deletion of the *lsrB* gene that encodes the only known AI-2 receptor in *S.* Typhimurium[26] did not affect its ability to form a biofilm, whereas the double mutant Δ*lsrB*Δ*luxS* showed significantly decreased biofilm formation compared with the Δ*lsrB* mutant (Fig. 1a). Moreover, deletion of *luxS* led to enhanced swimming motility in *S.* Typhimurium with or without the native *lsrB* gene (Supplementary Fig. 1b and Fig. 1b). These results suggest that AI-2 could play a role in the motile-sessile transition independent of its receptor LsrB.

Given that a recent study has found that dCache_1 domain-containing AI-2 receptors are widely distributed in bacteria[7], we investigated whether this type of AI-2 receptor is present in *S.* Typhimurium. Domain annotations of all protein sequences of *S.* Typhimurium by hmmscan program in HMMER (https://www.ebi.ac.uk/Tools/hmmer/search/hmmscan) showed that none of the protein sequences has the dCache_1 domain model (PF02743) as the best hit. Nevertheless, periplasmic ligand-binding domains (LBDs) of five transmembrane proteins DpiB (STM0625), YeaJ (STM1283), YedQ (STM1987), DcuS (STM4304), and CreC (STM4589) were found to hit the dCache_1 domain model with an E-value <1E-3. We thus examined whether these LBDs have the capacity to bind AI-2. In the *Vibrio harveyi* MM32 reporter assay, AI-2 binding activity was observed only for the LBD of YeaJ, but not for the LBDs of the other 4 proteins (Fig. 1c). Furthermore, the binding analysis by isothermal titration calorimetry (ITC) showed that the YeaJ-LBD binds AI-2 with a disassociation constant ($K_d$) value of $0.15 \pm 0.02\,\mu M$ (Fig. 1d). These results indicate that AI-2 is a high-affinity ligand for YeaJ.

YeaJ has been shown to be an active DGC that is involved in the regulation of motile-sessile transition in *E. coli* and *S.* Typhimurium[29,30]. By the in vitro DGC activity assay, we found that DPD/AI-2 stimulates the activity of YeaJ in synthesizing c-di-GMP (Fig. 1e and Supplementary Fig. 2). Consistent with this finding, when *S.* Typhimurium strains were cultured to the mid-exponential phase where the extracellular AI-2 activity reached the maximal level in the wild-type strain[31] (Supplementary Fig. 3a), intracellular c-di-GMP level in the Δ*luxS* mutant was significantly lower than that in the wildtype (Fig. 1f). Such reduction was partially restored by complementation with a plasmid encoding *luxS* (Fig. 1f), whereas the exogenous addition of DPD/AI-2 in cultures of Δ*luxS* resulted in a significant increase in intracellular c-di-GMP concentration (Fig. 1g). In contrast, deletion of *luxS* in the Δ*yeaJ* mutant did not lead to significant changes in the intracellular level of c-di-GMP (Supplementary Fig. 4), while the addition of DPD/AI-2 did not increase intracellular c-di-GMP concentration in Δ*yeaJ*Δ*luxS* (Fig. 1g). Consistent with previous studies[29,30], Δ*yeaJ* showed reduced biofilm formation but enhanced motility compared with the wildtype, whereas no significant difference in these phenotypes was observed between Δ*yeaJ*Δ*luxS* and Δ*yeaJ* (Supplementary Fig. 5a, b), suggesting that AI-2 regulates biofilm formation and motility via YeaJ. Collectively, these results indicate that AI-2 positively regulates intracellular c-di-GMP levels in *S.* Typhimurium by acting as an activator of YeaJ.

### YeaJ homologs that sense AI-2 are also present in *E. coli* and other members of *Enterobacterales*

While matching the dCache_1 domain model at a less stringent E-value (1.8E-4), the LBD of YeaJ had the GAPES1 model (PF17155) as the best hit (E-value = 4.5E-153) in hmmscan searches (Fig. 2a). BLASTP searching of the National Center for Biotechnology Information (NCBI) non-redundant protein database followed by domain predictions using InterProScan 5 against the Pfam database showed that YeaJ homologs that possess a putative N-terminal GAPES1 domain and a putative C-terminal GGDEF domain are mainly distributed in the order *Enterobacterales*, including members of the families *Enterobacteriaceae*, *Pectobacteriaceae* and *Hafniaceae* (Supplementary Data 1 and Supplementary Fig. 6). To examine the ability of the GAPES1 domains of these YeaJ homologs to bind AI-2, we randomly selected one YeaJ homolog from each genus and prepared recombinant His₆-GAPES1 proteins from the *luxS*⁺ *E. coli* strain. In the *V. harveyi* MM32 reporter assay, AI-2 binding activity was observed for the GAPES1 domains of YeaJ homologs from enterohemorrhagic *Escherichia coli* (EHEC) O157:H7 and 15 species from other genera (Fig. 2b). In contrast, no such activity was detected in the GAPES1 domains of two YeaJ homologs (KMK13526 and WP_034494714) from two species belonging to the genera *Pluralibacter* and *Buttiauxella* (Fig. 2b). Binding analysis by ITC showed that the GAPES1 domain of the YeaJ homolog from

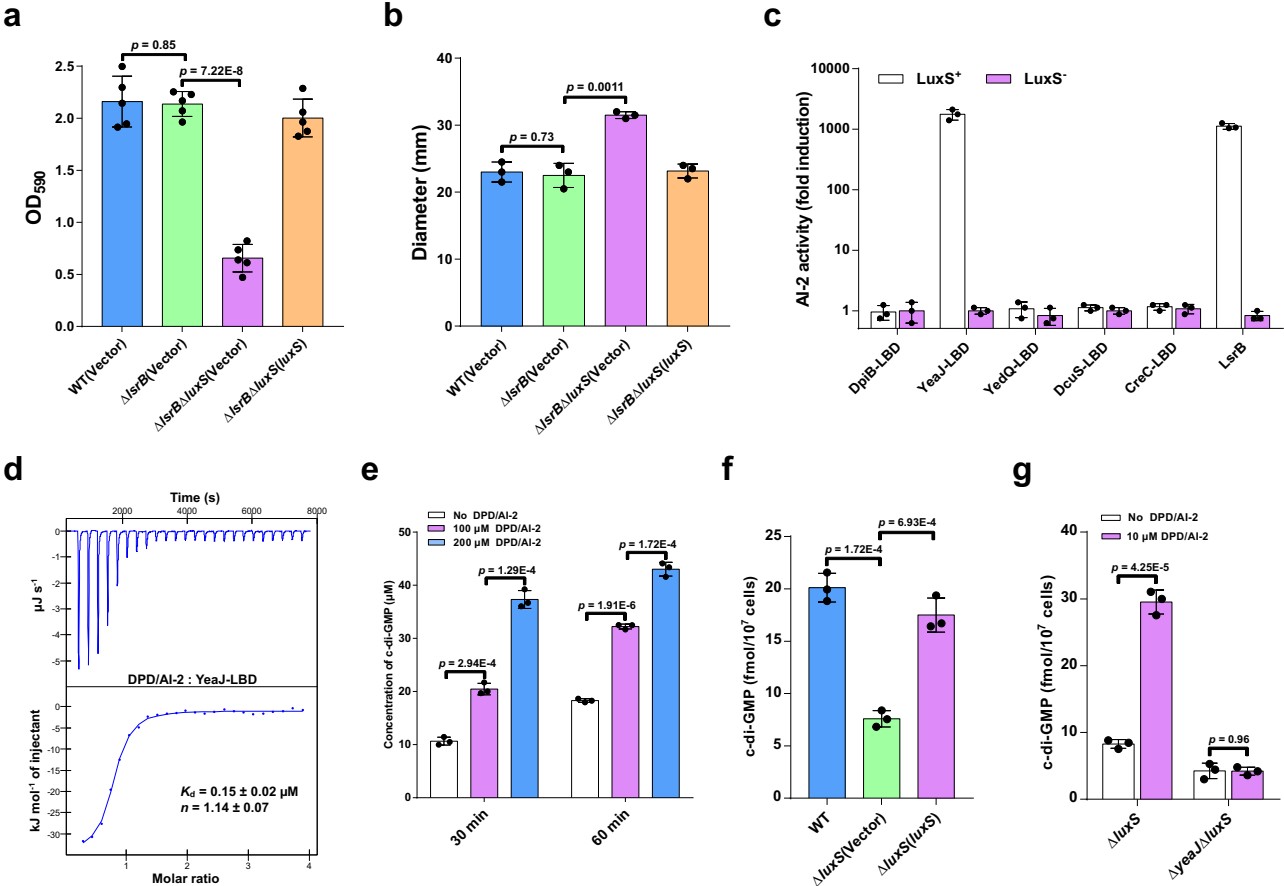

**Fig. 1 | AI-2 increases intracellular c-di-GMP levels by directly engaging the DGC YeaJ in *S.* Typhimurium. a** AI-2 regulates biofilm formation in *S.* Typhimurium independent of LsrB. Biofilms were stained with crystal violet and quantified using optical density measurement. Data were mean ± s.e.m. of five independent experiments. **b** AI-2 regulates the swimming motility of *S.* Typhimurium independent of LsrB. Data were mean ± s.e.m. of three independent experiments. **c** YeaJ-LBD is capable of retaining AI-2. Bioluminescence in *V. harveyi* MM32 (*luxN⁻*, *luxS⁻*) was induced by the addition of ligands released from purified proteins expressed in a *luxS⁺* or *luxS⁻ E. coli* strain. LsrB from *S.* Typhimurium was used as a positive control. AI-2 activity is reported as fold induction relative to the light production induced by a buffer control. Data were mean ± s.e.m. of three independent experiments. **d** ITC assays for the specific interaction between YeaJ-LBD and AI-2. The data shown are representative of three independent experiments with similar results. The $K_d$ and complex stoichiometry (*n*) values were presented as mean ± s.d. of three independent experiments. **e** AI-2 stimulates the DGC activity of YeaJ in vitro. Data were mean ± s.e.m. of three independent experiments. **f** Liquid chromatography-tandem mass spectrometry (LC-MS/MS) measurements of cellular levels of c-di-GMP in *S.* Typhimurium strains. The bacterial cultures grown in LB broth at 37 °C with shaking to an $OD_{600}$ of 1.3 were subjected to nucleotide extractions. Data represent mean ± s.d. from three biological replicates. **g** AI-2 increases cellular c-di-GMP concentration via YeaJ. The mutants Δ*luxS* and Δ*yeaJ*Δ*luxS* grown in LB medium to an $OD_{600}$ of 1.3 were induced by 10 μM DPD/AI-2 or a buffer control for 30 min, and then bacterial cells were collected to extract nucleotides for LC-MS/MS analysis. Data shown are mean ± s.d. of three biological replicates. **a**, **b**, **e**–**g** Statistical significance was evaluated using a two-tailed unpaired Student's *t*-test and $p < 0.05$ was considered statistically significant. WT wildtype. Source data are provided as a Source Data file.

EHEC O157:H7 binds AI-2 with high affinity (Supplementary Fig. 7a). These data suggest that GAPES1 is a type of extracytoplasmic sensor recognizing the AI-2 signal and YeaJ homologs whose DGC activities can be regulated by AI-2 are widespread among members of *Enterobacterales*.

By amino acid sequence alignment of the GAPES1 domains that have been tested for AI-2 binding activity, we found that two residues corresponding to Y210 and D239 of YeaJ are conserved in AI-2-binding GAPES1 domains but not in the two GAPES1 domains with no AI-2 binding activity (Fig. 2c). Mutations in each of these two residues resulted in a marked reduction in AI-2 binding affinity for YeaJ-LBD (Figs. 1d, 2d). Furthermore, mutations of both non-conserved residues to conserved residues within KMK13526-LBD (H216Y/Q245D) and WP_034494714-LBD (Q208Y/E237D) increased their AI-2 binding affinity to levels (0.16–0.21 μM) (Supplementary Fig. 7b–e) that were comparable to that of YeaJ-LBD (Fig. 1d). These results suggest that the two highly conserved positions of the GAPES1 domains corresponding to Y210 and D239 of YeaJ may be key residues for AI-2 binding.

## AI-2 negatively controls the T3SS-1 and attenuates the virulence of *S.* Typhimurium in infection via YeaJ

High levels of c-di-GMP in *S.* Typhimurium have been shown to reduce the secretion of T3SS-1 effectors as well as invasion of epithelial cells[23–25]. As expected, when grown under *Salmonella* pathogenicity island 1 (SPI-1) inducing conditions[32] to the mid-exponential phase (Supplementary Fig. 3b), deletion of *luxS* or *yeaJ* significantly promoted intracellular accumulation and secretion of the T3SS-1 effectors SipB and SopB (Fig. 3a). Such induction was abolished by the expression of *luxS* and *yeaJ* in the corresponding mutants (Fig. 3a). However, the double mutant Δ*yeaJ*Δ*luxS* produced and secreted SipB and SopB at levels similar to those by Δ*yeaJ* (Fig. 3a). These results suggest that AI-2 negatively regulates the production and secretion of T3SS-1 effectors through YeaJ.

We then investigated whether the rise in protein levels of SipB and SopB in the mutants Δ*luxS* and Δ*yeaJ* is due to increased expression of *sipB* and *sopB* at the transcriptional levels. Quantitative real-time PCR (qRT-PCR) analysis showed that the mRNA levels of *sipB* and *sopB* were

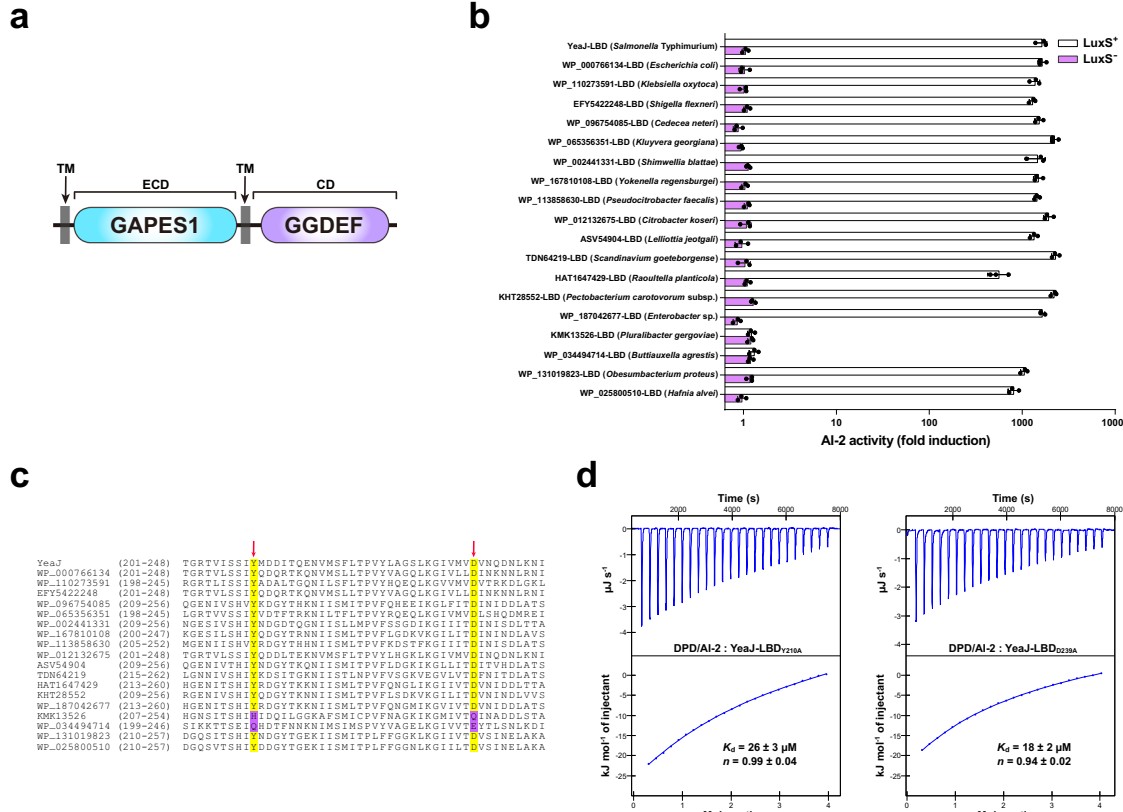

**Fig. 2 | Widespread occurrence of GAPES1 domain-containing YeaJ homologs capable of sensing AI-2 in the order *Enterobacterales*. a** Schematic illustrating the predicted domain organization of YeaJ homologs. Protein sequences were analyzed using hmmscan program against the Pfam 34.0 database. TM transmembrane domain, ECD extracytoplasmic domain, CD cytoplasmic domain. **b** GAPES1 domains from YeaJ homologs in bacterial species belonging to the *Enterobacteriaceae*, *Pectobacteriaceae*, and *Hafniaceae* families are capable of retaining AI-2. GAPES1 domains from YeaJ homologs were expressed and purified as His$_6$ fusion proteins in a *luxS*+ or *luxS*- *E. coli* strain. Bioluminescence in *V. harveyi* strain MM32 was measured following the addition of a buffer control or ligands released from the purified proteins upon denaturing by heating. YeaJ-LBD was used as a positive control. AI-2 activity is presented as mean ± s.e.m. of three independent experiments. The NCBI accession numbers for YeaJ homologs are provided and the

bacterial species to which they belong are given in parentheses. **c** AI-2-binding GAPES1 domains of YeaJ homologs harbor two conserved residues corresponding to Y210 and D239 of YeaJ. GAPES1 domains of YeaJ and its homologs that have been tested for AI-2 binding activity were aligned using ClustalW embedded in MEGA7 software. Two positions corresponding to Y210 and D239 of YeaJ that may be involved in AI-2 binding are labeled with red arrows. The conserved residues in the two positions are highlighted in yellow and non-conserved residues are highlighted in purple. The NCBI accession number for each YeaJ homolog is included. **d** Isotherms representing binding of two mutants of YeaJ-LBD (Y210A or D239A) with DPD/AI-2. The binding affinity was determined using ITC. The data shown were one representative of three independent experiments with similar results. The $K_d$ and complex stoichiometry (*n*) are presented as mean ± s.d. of three independent experiments. Source data are provided as a Source Data file.

significantly higher in Δ*luxS* and Δ*yeaJ* compared to the wild-type and complemented strains (Fig. 3b). Similar observations were made for the expression of *sopE2* (Fig. 3b), which encodes a T3SS-1 effector whose secretion is also regulated by c-di-GMP signaling[23]. Moreover, promoter-reporter assays showed that deletion of *luxS* or *yeaJ* led to significantly increased promoter activities of *sopB*, *sopE2*, and the *sicAsipBCDA* operon (Fig. 3c). These results indicate that AI-2-induced elevated c-di-GMP inhibits transcription of *sopB*, *sopE2*, and the *sicAsipBCDA* operon.

We further investigated the ability of *S.* Typhimurium strains to adhere to and invade human colonic epithelial Caco-2 cells. In contrast to the wild-type parent strain, mutants lacking *luxS* or *yeaJ* showed slightly enhanced adherence to (Fig. 3d), and significantly increased invasion of Caco-2 cells (Fig. 3e). Complementation returned their adherence and invasion ability to wild-type levels (Fig. 3d, e). However, in contrast to our results, secretion of T3SS-1 effectors and the ability to invade epithelial cells were not altered in Δ*luxS* compared to the wildtype in a previous study by ref. 33. We note that different culture conditions were used for both assays in the two studies. In our study, both assays were performed using cultures grown in a modified LB medium containing 0.3 M NaCl without agitation (a condition for

induction of the T3SS-1 encoded on SPI-1[32,34]) to mid-exponential phase, when the AI-2 activity in the culture supernatant of the wild-type strain was maximal (Supplementary Fig. 3b), while the study by Perrett et al.[33]. used shaking cultures in normal LB medium with an OD$_{600}$ of 1.0 for T3SS-1 secretion assays and in the late log phase for invasion assays. We also found no differences between the wildtype and Δ*luxS* with respect to their ability to invade epithelial cells in the conditions that ref. 33 used (Supplementary Fig. 8). Thus, the discrepancy observed in LuxS regulation of the T3SS-1 and invasion of epithelial cells can be explained by the use of different culture conditions.

To further investigate whether deletion of *luxS* or *yeaJ* affects intestinal colonization after infection by a natural route, we performed competitive oral infections of streptomycin-treated BALB/c mice with an equal mixture of wild-type and mutant strains of *S.* Typhimurium. Competition assays showed that Δ*luxS*, Δ*yeaJ*, and Δ*yeaJ*Δ*luxS* outcompeted the wildtype ~2 to 3-fold, ~8 to 27-fold, and ~8 to 28-fold, respectively (Fig. 3f). Competitive indexes between Δ*luxS* and the wildtype, although drastically lower than those between Δ*yeaJ* and the wildtype, are statistically different from a control competition assay between two derivatives of the wild-type SL1344 carrying the kanamycin-resistant pKT100 and the chloramphenicol-resistant

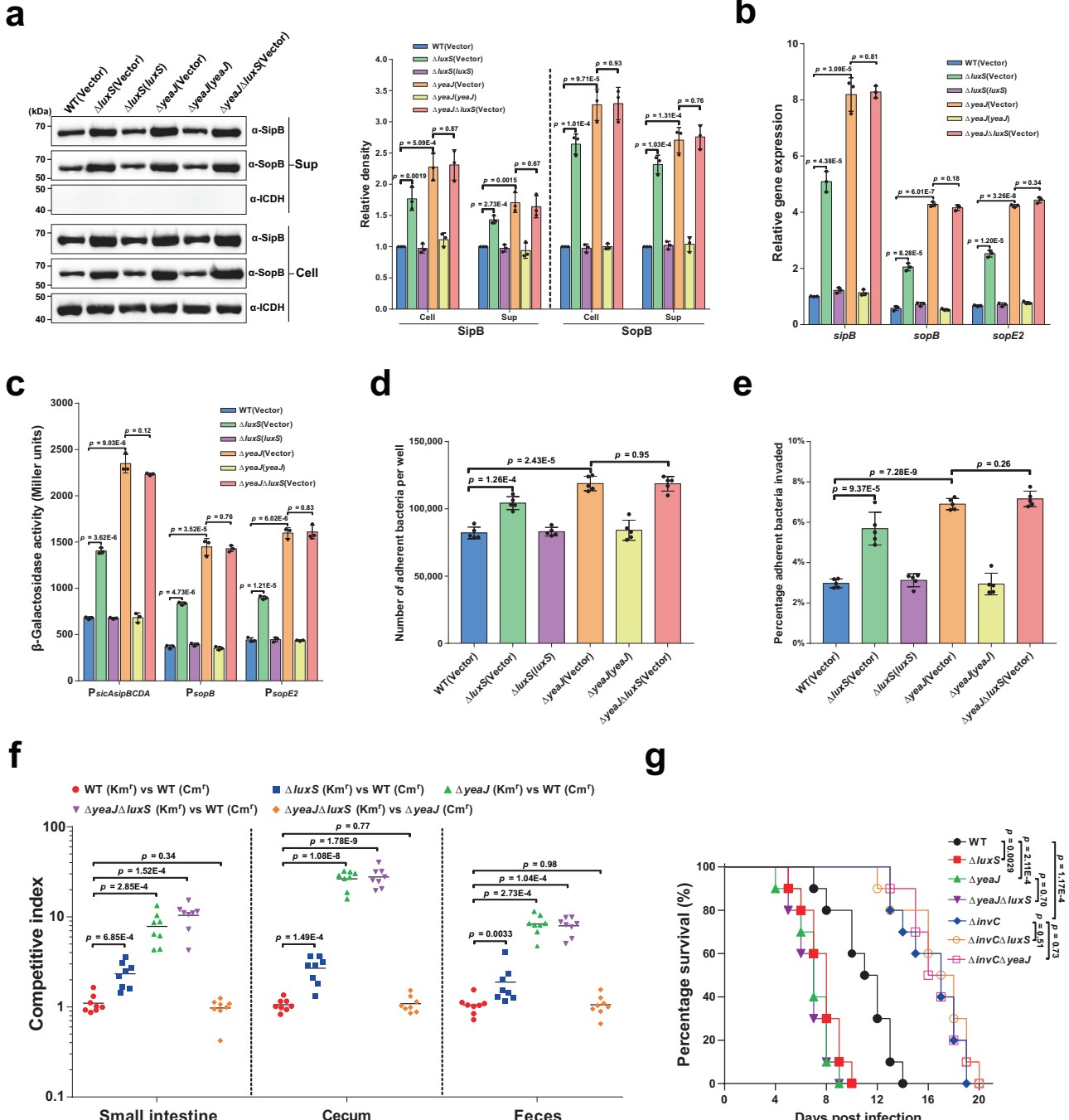

**Fig. 3 | AI-2 represses the T3SS-1 and negatively regulates the virulence of *S.* Typhimurium via YeaJ. a** AI-2 negatively regulates intracellular accumulation and secretion of SipB and SopB via YeaJ. Cell pellet (Cell) and concentrated supernatant (Sup) were probed for SipB and SopB by western blot analysis. Isocitrate dehydrogenase (ICDH) was probed as a loading control. The blots shown are representative of three independent experiments with similar results. The band intensities were quantified by scanning densitometry using ImageJ (NIH, USA), normalized to intracellular ICDH, and presented as values relative to that of the wildtype (mean ± s.d.; *n* = 3 independent experiments). **b** mRNA levels of T3SS-1 genes were determined by qRT-PCR analyses. Expression was normalized to 16 S rRNA and reported as fold change relative to that of the *sipB* gene of the wildtype. Data were mean ± s.e.m. of three independent experiments. **c** The promoter activities of T3SS-1 genes measured using β-galactosidase activity assays (mean ± s.e.m.; *n* = 3 independent experiments). **d**, **e**, Adherence to (**d**) and invasion of (**e**)

Caco-2 cells by *S.* Typhimurium strains. Data were mean ± s.e.m. of five independent experiments. **f** Six-week-old female BALB/c mice were infected orally with a 1:1 mixture of two *S.* Typhimurium strains carrying kanamycin-resistant (Km$^r$) pKT100 and chloramphenicol-resistant (Cm$^r$) pBBR1MCS1, respectively. A competitive index (CI) was calculated as the ratio of the test strain carrying pKT100 versus (vs) the control strain carrying pBBR1MCS1 recovered from mice. Horizontal lines represent the geometric mean CI value for each group (*n* = 8 mice per group). **g** Deletion of *luxS* or *yeaJ* leads to enhanced virulence of *S.* Typhimurium in mice. Six-week-old female BALB/c mice were infected orally with each *S.* Typhimurium strain and survival was monitored daily. Data were illustrated as a percentage of mice survival (*n* = 10 mice per group). Statistical significance was evaluated by two-tailed unpaired Student's *t*-test (**a**–**e**), two-tailed Mann–Whitney *U*-test (**f**), or Log-rank (Mantel–Cox) test (**g**). *P* values <0.05 indicate significant differences. Source data are provided as a Source Data file.

pBBR1MCS1, respectively (Fig. 3f). In contrast, when $\Delta yeaJ\Delta luxS$ competed against $\Delta yeaJ$, these two mutants were recovered at similar levels in the small intestine, cecum, and feces (Fig. 3f). Similar results were also observed when competitions were conducted using the same strains with swapped antibiotic markers (Supplementary Fig. 9). These data suggest that AI-2 negatively regulates *S*. Typhimurium intestinal colonization via YeaJ.

We also evaluated the lethality of *S*. Typhimurium strains in BALB/c mice. In an oral infection model, $\Delta luxS$ and $\Delta yeaJ$ led to significantly increased mouse mortality compared to the wild-type strain, whereas infections with the mutants $\Delta yeaJ\Delta luxS$ and $\Delta yeaJ$ produced similar mortality (Fig. 3g). Consistent with the role of SPI-1 in intestinal infection[35,36], deletion of the gene that encodes the T3SS-1 ATPase InvC resulted in decreased mortality of mice after oral challenge (Fig. 3g). Moreover, mice infected with $\Delta invC\Delta luxS$ and $\Delta invC\Delta yeaJ$ showed similar mortality compared to those infected with $\Delta invC$ (Fig. 3g), indicating that AI-2-mediated c-di-GMP signaling regulates the virulence of *S*. Typhimurium via T3SS-1. However, deletion of *luxS* or *yeaJ* did not affect the virulence of *S*. Typhimurium after intraperitoneal inoculation (Supplementary Fig. 10), suggesting that AI-2-induced repression of the T3SS-1 via YeaJ has no major impact on systemic infection. Together, these results indicate that AI-2 exerts a negative regulatory effect on *S*. Typhimurium virulence during intestinal infection by modulating the function of the T3SS-1 via YeaJ.

### Bile salts stimulate the DGC activity of YedQ to repress the T3SS-1 in *S*. Typhimurium

Bile, a major host-produced heterogeneous mixture of compounds encountered by bacteria in the small intestine, was previously shown to repress the expression of invasion-related genes within SPI-1 in *S*. Typhimurium[37,38], but the mechanism of such regulation is poorly understood. While bile has been reported to increase intracellular c-di-GMP levels in *V. cholerae*[10,14], our results showed that AI-2 induces transcriptional repression of T3SS-1 genes via c-di-GMP signaling (Fig. 3b, c), leading us to speculate that bile salts may modulate intracellular c-di-GMP levels to repress the T3SS-1 in *S*. Typhimurium. Indeed, when strain SL1344 was stimulated by porcine bile salts at a concentration (0.05%, w/v) comparable to that physiologically occurring in intestinal contents[39], the cellular concentration of c-di-GMP increased approximately fivefold (Fig. 4a). As expected, the promoter activities of *sopB*, *sopE2*, and *sicAsipBCDA* in the wild-type strain were significantly repressed following exposure to 0.05% bile salts (Fig. 4b). These results suggest that bile salts repress T3SS-1 gene expression via increasing intracellular c-di-GMP levels in *S*. Typhimurium.

Consistent with the known role of c-di-GMP in promoting biofilm formation[23,29], the addition of 0.05% bile salts in cultures of strain SL1344 resulted in a significant increase in biofilm production (Fig. 4c). We further tested nine individual components of bile salts to determine their contributions to biofilm formation. Intriguingly, the addition of taurocholate and taurodeoxycholate significantly stimulated biofilm formation, while the remaining seven components of bile have no such effect (Fig. 4c). To identify CMEs that robustly respond to bile salts, we deleted each of the 17 genes encoding predicted CMEs[6] and examined the ability of the mutants to form biofilms in response to bile salts and the individual bile components taurocholate and taurodeoxycholate. Whereas most of these mutations did not affect the response of *S*. Typhimurium to bile salts, the deletion of *yedQ* completely abrogated the bile-induced enhancement of biofilm formation (Fig. 4d). Moreover, the inclusion of 1 µM taurocholate or taurodeoxycholate in cultures of the wild-type strain resulted in a significant increase in intracellular c-di-GMP levels, while such induction was completely abolished in $\Delta yedQ$ (Fig. 4e). Complementation of the mutant with a plasmid derived copy of *yedQ* restored c-di-GMP modulation in response to taurocholate and taurodeoxycholate (Fig. 4e). These observations indicate that the bile components taurocholate

and taurodeoxycholate stimulate an increase in intracellular c-di-GMP concentrations via the DGC YedQ.

We further examined whether the LBD of YedQ directly interacts with taurocholate and taurodeoxycholate. Binding analysis by ITC showed that taurocholate and taurodeoxycholate bind to YedQ-LBD with $K_d$ values of $0.17 \pm 0.03$ µM and $0.14 \pm 0.02$ µM, respectively (Fig. 4f). We then predicted the 3D structure of YedQ-LBD by Alphafold2[40] and performed a docking simulation to analyze the interaction between YedQ-LBD and taurocholate. The best docking conformation obtained by AutoDock Vina 1.1.2[41] suggests that taurocholate is inserted into a big cavity of YedQ-LBD (Fig. 4g). This conformation suggests that taurocholate makes close contact with D233, K236, Q264, L288, L289, D291, E295 and Q297 in the cavity (Fig. 4h). In support of this binding model, mutations in D233, K236, Q264, L288, and L289 drastically reduced the binding affinity of YedQ-LBD for taurocholate (Fig. 4i and Supplementary Fig. 11). Furthermore, taurocholate and taurodeoxycholate were able to induce the DGC activity of YedQ in c-di-GMP synthesis (Fig. 4j and Supplementary Fig. 12). These results indicate that taurocholate and taurodeoxycholate can induce intracellular accumulation of c-di-GMP in S. Typhimurium by directly engaging YedQ.

Next, we tested whether the expression of T3SS-1 genes is affected by the presence of taurocholate and taurodeoxycholate. The addition of 1 µM taurocholate or taurodeoxycholate strongly repressed the promoter activities of *sopB*, *sopE2*, and *sicAsipBCDA* in the wild-type strain, whereas this effect was completely abolished in the $\Delta yedQ$ mutant (Fig. 4k), indicating that taurocholate and taurodeoxycholate repress T3SS-1 gene expression via YedQ. Taken together, our data reveal that bile components taurocholate and taurodeoxycholate stimulate the DGC activity of YedQ to repress T3SS-1 gene expression in *S*. Typhimurium.

### C-di-GMP affects the expression and secretion of T3SS-1 effectors through binding to the T3SS-1 chaperone SicA

Based on the above observations, we speculated that there may exist a yet unidentified c-di-GMP-binding effector that regulates T3SS-1 gene expression at the transcriptional level. It was previously shown that the transcription factor InvF in complex with the T3SS-1 chaperone SicA directly activates the expression of SipB and SopB[35,42]. However, the mRNA levels of *invF* as well as its promoter activity were similar among the wildtype, mutants $\Delta luxS$ and $\Delta yeaJ$, and the corresponding complemented strains (Supplementary Fig. 13), eliminating the possibility that an increase in *invF* expression results in elevated expression and production of SipB and SopB. A range of transcription factors have previously been shown to bind to c-di-GMP, leading to altered DNA binding capacity and, thus, changes in the expression of downstream target genes[1,16]. We thus examined whether InvF is a c-di-GMP effector that regulates downstream gene expression in response to this ligand. However, ITC analysis showed that InvF does not bind c-di-GMP (Supplementary Fig. 14a). Of note, a number of small proteins were reported to act as c-di-GMP-binding adapters that regulate the catalytic or binding properties of their protein partners in a c-di-GMP-dependent manner[20,21], thus raising another possibility that SicA is a c-di-GMP sensor. Intriguingly, binding analysis by ITC showed that SicA binds c-di-GMP at a 1:1 stoichiometry with a $K_d$ of $0.21 \pm 0.09$ µM (Fig. 5a), which is comparable to previously reported $K_d$ values for several well-established c-di-GMP receptors[18-21]. By contrast, no binding of c-di-AMP or cGMP to SicA was detected under the same experimental conditions (Supplementary Fig. 14b, c). These results indicate that c-di-GMP directly and specifically binds to SicA.

We further performed co-immunoprecipitation (co-IP) assays to investigate how c-di-GMP affects the interaction between SicA with InvF. We constructed two plasmids to express C-terminal His$_6$-tagged InvF and hemagglutinin (HA)-tagged SicA, respectively, and found that the co-IP of InvF-His$_6$ with SicA-HA was impaired by c-di-GMP in a

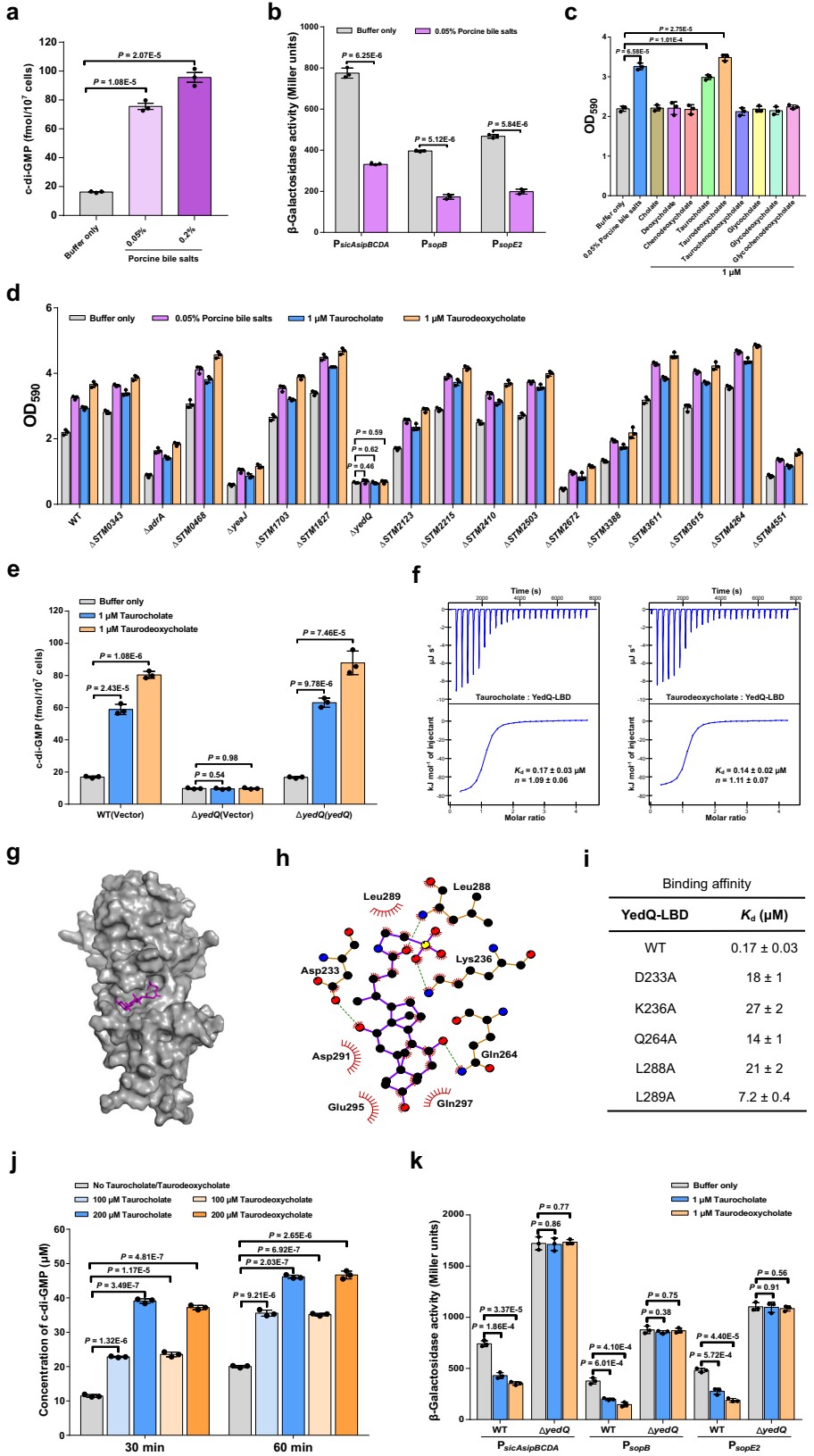

dose-dependent fashion, but not by high concentrations of c-di-AMP or cGMP (Fig. 5b). We also performed electrophoretic mobility shift assays (EMSAs) to further evaluate the effect of c-di-GMP on the interactions of the InvF/SicA complex with its target promoters. The EMSA results showed that the InvF/SicA complex specifically binds to the promoter sequences of *sopB*, *sopE2*, and the *sicAsipBCDA* operon

(Supplementary Fig. 15a–c). The addition of increasing concentrations of c-di-GMP reduced or even abrogated the formation of InvF/SicA-DNA complexes, whereas the inclusion of c-di-AMP or cGMP at a dose corresponding to the highest level of c-di-GMP did not result in an observable difference in the InvF/SicA-DNA binding (Fig. 5c–e). Collectively, these data suggest that the binding of c-di-GMP to SicA

**Fig. 4 | Bile components taurocholate and taurodeoxycholate repress the expression of T3SS-1 genes via stimulating the c-di-GMP synthase activity of YedQ. a** Bile salts stimulate an increase in intracellular c-di-GMP concentrations. Data were shown as mean ± s.d. of three biological replicates. **b** Bile salts inhibit the promoter activities of T3SS-1 genes. The promoter activity was determined by quantifying β-galactosidase activity. Data were mean ± s.e.m. of three independent experiments. **c** Bile salts and the individual bile components taurocholate and taurodeoxycholate stimulate biofilm formation in *S.* Typhimurium. Data were mean ± s.e.m. of three independent experiments. **d** Bile-induced enhancement of biofilm formation in *S.* Typhimurium requires YedQ. Data were presented as mean ± s.e.m. of three independent experiments. **e** An increase in intracellular c-di-GMP concentrations in response to taurocholate and taurodeoxycholate requires YedQ. Data were mean ± s.d. of three biological replicates. **f** Taurocholate and taurodeoxycholate bind to the LBD of YedQ with high affinity. ITC data shown are one representative of three independent experiments with similar results. $K_d$ and complex stoichiometry (*n*) are presented as mean ± s.d. of three independent experiments. **g** Surface representation of the structural model of YedQ-LBD in complex with taurocholate, prepared using PyMOL. Taurocholate is shown as purple sticks. **h** Schematic of the predicted contacts between taurocholate and YedQ-LBD from the taurocholate-binding conformation. Potential hydrogen bonds are indicated as green dashed lines. **i** Binding of taurocholate to YedQ-LBD and its mutants. The binding affinity was measured by ITC. The $K_d$ values are presented as mean ± s.d. of three independent experiments. **j** Taurocholate and taurodeoxycholate enhance the DGC activity of YedQ in vitro. Data represent mean ± s.e.m. of three independent experiments. **K** The promoter activities of T3SS-1 genes were inhibited by taurocholate and taurodeoxycholate in the wild-type strain, but not in Δ*yedQ*. Data were mean ± s.e.m. of three independent experiments. **a–e, j, K** *P* values were determined using the two-tailed unpaired Student's *t*-test. A *p* value less than 0.05 was considered to be statistically significant. Source data are provided as a Source Data file.

inhibits the formation of the InvF/SicA complex, thus resulting in reduced binding of the complex to its target promoters.

SipB and SipC are T3SS-1 translocators that also act as effector proteins[35,43,44]. In addition to acting as a co-activator of InvF, SicA also functions to partition and stabilize SipB and SipC by binding directly to them[45]. Immunoprecipitation experiments revealed that SicA-HA co-precipitated both SipB-His₆ and SipC-His₆, while c-di-GMP decreased the co-IP of SipB-His₆ and SipC-His₆ with SicA-HA in a dose-dependent fashion (Fig. 5f, g). We thus hypothesized that c-di-GMP could also affect the stability and secretion of SipB and SipC at the post-translational level by inhibiting the binding of SicA to SipB and SipC. To experimentally test this hypothesis, we replaced the promoter of the *sicAsipBCDA* operon in the wild-type and mutant strains with the *invF* promoter, whose activity is not regulated by c-di-GMP signaling (Supplementary Fig. 13). After promoter replacement, the expression of *sopB* and *sopE2* with their own promoters was still significantly upregulated in Δ*luxS* and Δ*yeaJ* but dramatically reduced in Δ*sicA* when compared with the wildtype, whereas the expression of *sipB* and *sipC* under the control of the *invF* promoter was not affected by deletion of *luxS*, *yeaJ*, or *sicA* (Supplementary Fig. 16). In contrast, western blot analysis showed that intracellular accumulation and secretion of SipB and SipC were increased in Δ*luxS* and Δ*yeaJ* compared to the wildtype, but reduced to very low levels in Δ*sicA* (Fig. 5h). These results support that changes in c-di-GMP concentration also play a role in the stability and secretion of SipB and SipC through targeting their chaperone SicA.

To study how SicA interacts with c-di-GMP, we constructed a homology model of SicA based on IpgC from *Shigella flexneri* (PDB ID: 3GYZ; https://www.rcsb.org/structure/3GYZ) using the Phyre2 server[46]. Potential ligand-binding sites were predicted using POCASA 1.1[47] and the molecular docking was performed by AutoDock Vina 1.1.2[41]. The conformation with the lowest binding energy of −8.1 kcal mol⁻¹ (Fig. 5i) suggests that c-di-GMP makes close contact with T25, K27, D28, Q34, D67, Y69, N70, P71, and D72 of SicA (Fig. 5j). Mutation of K27, D28, Q34, D67, or N70 resulted in a marked reduction in the c-di-GMP-binding affinity for SicA (Fig. 5k and Supplementary Fig. 17), indicating that these residues are directly involved in c-di-GMP binding. In addition, protein-protein docking analysis by Cluspro 2.0[48] suggested that InvF, SipB, and SipC have partially overlapping interaction surfaces on SicA, while the interaction surfaces of SicA with SipB and SipC, but not with InvF, partially overlap with the c-di-GMP-binding site (Supplementary Fig. 18 and Fig. 5j). In addition to SipB and SipC, the binding of InvF to SicA was also disturbed by c-di-GMP binding (Fig. 5b), suggesting that c-di-GMP allosterically regulates SicA. Among residues of SicA that make contact with c-di-GMP, K27, D28, Q34, and D67, but not N70, were predicted to participate in interactions with its protein partners SipB and SipC (Supplementary Fig. 18). Indeed, the K27A variant showed a 19-fold lower binding affinity to SipB compared with wild-type SicA (Supplementary Fig. 19a, b). In contrast, the N70A mutation of SicA did not affect its binding affinities for InvF, SipB, and SipC (Supplementary

Fig. 19a, c–g). Moreover, changing N70 to alanine did not affect its ability to co-immunoprecipitate InvF-His₆, SipB-His₆, and SipC-His₆ without the addition of c-di-GMP, whereas high concentrations of c-di-GMP failed to impair co-IP of InvF-His₆, SipB-His₆, and SipC-His₆ with SicA_{N70A}-HA (Supplementary Fig. 20). These results suggest that allosteric regulation of SicA by c-di-GMP is abrogated by an N70A mutation. Thus, changing N70 to alanine specifically impairs the binding of c-di-GMP but leaves its chaperone function unaffected.

When SicA_{N70A} was expressed at a level similar to that of wild-type SicA, the amounts of InvF, SipB, and SipC bound by SicA_{N70A} were much higher than those bound by wild-type SicA after the addition of the same concentrations of c-di-GMP (Supplementary Fig. 20), which can be attributed to the much lower c-di-GMP-binding affinity of SicA_{N70A} and thus less binding of SicA_{N70A} to c-di-GMP when compared with wild-type SicA. To determine the role of c-di-GMP on SicA activity in vivo, we replaced the wild-type *sicA* gene in the chromosome with *sicA*(N70A). While deletion of *sicA* significantly reduced the expression of *sipB*, *sopB*, and *sopE2*, the expression levels of these genes were drastically increased in the *sicA*(N70A) mutant compared to the wild-type strain (Fig. 5l), indicating that less binding of SicA_{N70A} to c-di-GMP but more binding of this chaperone to InvF leads to enhanced transcription of the target genes of InvF/SicA in the *sicA*(N70A) mutant. Consistent with previous findings that *sicA* is also a target gene of InvF/SicA[35,42], the expression level of the *sicA*(N70A) gene in the *sicA*(N70A) mutant was significantly higher than that of *sicA* in the wildtype (Fig. 5l). Furthermore, deletion of *luxS* or *yeaJ* in the *sicA*(N70A) mutant background did not alter the expression of the T3SS-1 genes (Fig. 5l), suggesting that SicA_{N70A} is unable to bind c-di-GMP at its physiological concentration. In the murine oral infection model, in contrast to the mutants Δ*luxS* and Δ*yeaJ*, the Δ*sicA* mutant led to significantly decreased mouse mortality compared to the wild-type strain (Fig. 5m). Complementation with plasmid-borne *sicA* restored the lethality of the Δ*sicA* mutant in mice to wild-type levels (Fig. 5m). By contrast, mice infected with the *sicA*(N70A) mutant showed significantly higher mortality than those infected with the wildtype, whereas infections with *sicA*(N70A) and its derivative mutants lacking *luxS* or *yeaJ* produced similar mortality (Fig. 5m). These in vivo observations indicate that the N70A mutation of the *sicA* gene in *S.* Typhimurium promotes the T3SS-1 activity but abolishes responses to c-di-GMP signaling.

Taken together, these in vitro and in vivo results suggest that c-di-GMP exerts its regulatory effects on T3SS-1 through binding to SicA, and elevated intracellular levels of c-di-GMP will lead to less binding of SicA to InvF, SipB, and SipC, thus downregulating the expression of the T3SS-1 genes as well as impairing the stability and secretion of SipB and SipC.

### c-di-GMP-binding SicA homologs are widely distributed among Gram-negative pathogenic bacteria

SicA of *S.* Typhimurium belongs to class II of T3SS chaperones and harbors three tandem tetratricopeptide repeat (TPR) motifs, which is

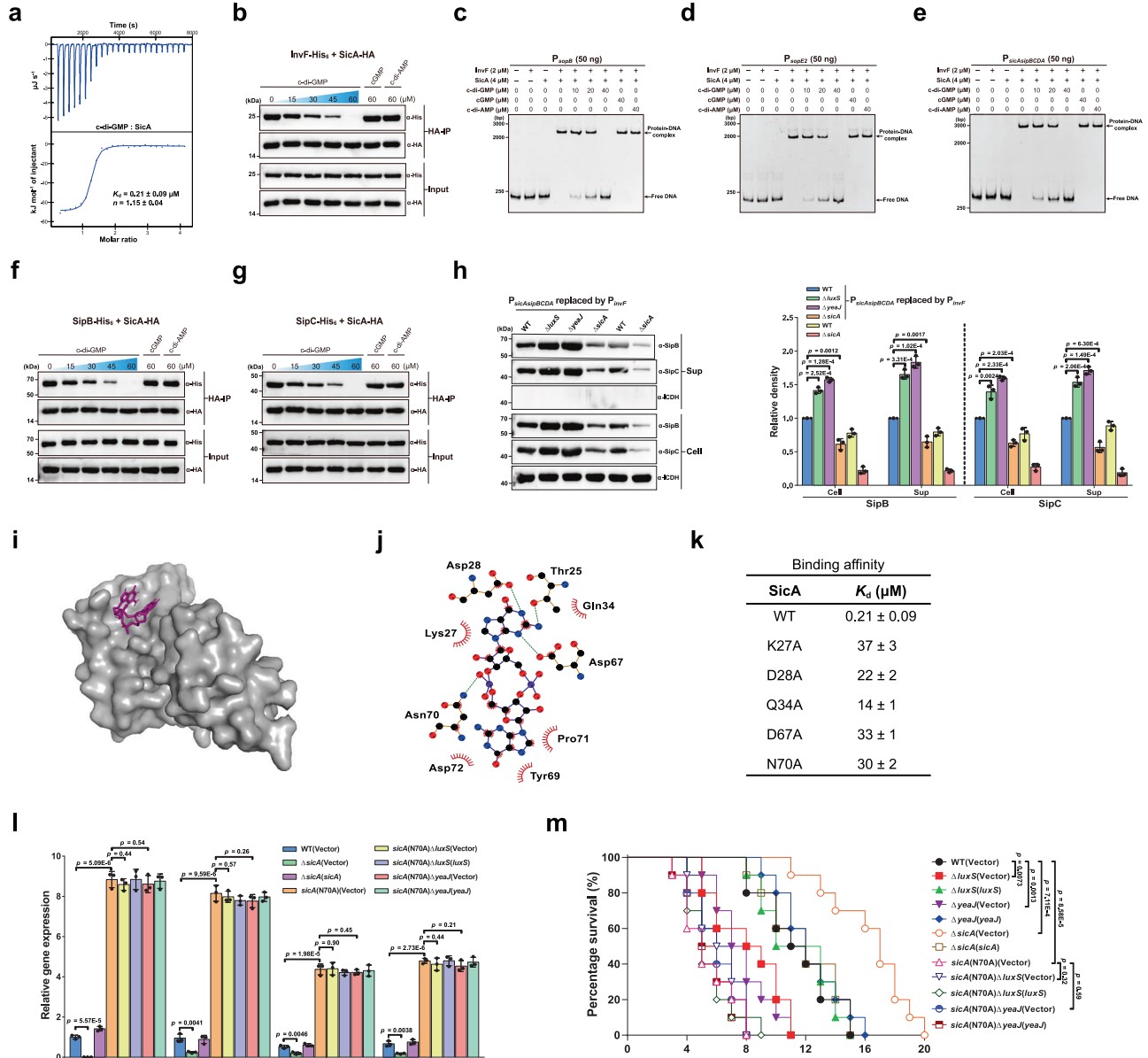

**Fig. 5 | Regulation of the expression and secretion of T3SS-1 effectors by c-di-GMP signaling is dependent on the binding of c-di-GMP to SicA. a** ITC analysis of c-di-GMP binding to SicA. Data shown are one representative of three independent experiments with similar results, with $K_d$ and complex stoichiometry ($n$) presented as mean ± s.d. **b** Co-IP of InvF-His$_6$ with SicA-HA is impaired by c-di-GMP, but not by cGMP or c-di-AMP. The immunoprecipitated proteins (HA-IP) and the total cell lysates (Input) were assessed by western blot analysis. **c–e** EMSAs for InvF/SicA binding to promoters of *sopB* (**c**), *sopE2* (**d**), and the *sicAsipBCDA* operon (**e**) in the presence and absence of nucleotides. **f, g** c-di-GMP decreased the Co-IP of SipB-His$_6$ (**f**) and SipC-His$_6$ (**g**) with SicA-HA in a dose-dependent fashion. **h** Western blot analysis of intracellular accumulation and secretion of SipB and SipC by strains with P$_{sicAsipBCDA}$ replaced by P$_{invF}$. Band intensities were presented as values relative to that of the wildtype with promoter replacement (mean ± s.d.; $n$ = 3 independent experiments). **i** Surface representation of the homology model of SicA in complex

with c-di-GMP. c-di-GMP is shown as purple sticks. **j** Schematic of the predicted contacts between c-di-GMP and SicA. Potential hydrogen bonds are indicated as green dashed lines. **k** Binding of c-di-GMP to SicA and its mutants, as measured by ITC ($K_d$ = mean ± s.d.; $n$ = 3 independent experiments). **l** qRT-PCR analyses of the mRNA levels of T3SS-1 genes. Expression was presented as values relative to that of *sicA* of the wildtype (mean ± s.e.m.; $n$ = 3 independent experiments). **m** Survival curves of 6-week-old female BALB/c mice infected orally with *S*. Typhimurium strains. Data were illustrated as a percentage of mice survival ($n$ = 10 mice per group). **b–h** Gels or blots shown are one representative of three independent experiments with similar results. Statistical significance was calculated using the Student's *t*-test (**h**, **l**) or Log-rank (Mantel–Cox) test (**m**). **h**, **l**, **m** Differences were considered statistically significant at $p$ < 0.05. Source data are provided as a Source Data file.

characteristic of the CesD/SycD/LcrH family of T3SS chaperones[49–51]. Homology searches using the BLASTP program against the NCBI non-redundant protein database revealed that the CesD/SycD/LcrH family of chaperones that shares >20% sequence identity with SicA (E-value cutoff of 1E-03) are widely distributed in Gram-negative bacteria, especially the phylum *Proteobacteria* (Supplementary Data 2 and Supplementary Fig. 21). To examine whether c-di-GMP binding is a

common feature of this family, T3SS chaperone proteins from several well-known pathogenic bacteria, including PcrH of *Pseudomonas aeruginosa*, IpgC of *S. flexneri* serotype 2a, SycD of *Yersinia enterocolitica*, CesD of EHEC O157:H7, BicA of *Burkholderia thailandensis* and VcrH of *Vibrio parahaemolyticus*, were expressed and purified as recombinant proteins from *E. coli* BL21(DE3) and their ability to bind c-di-GMP was assessed. Strikingly, while these 6 chaperones share 26.2–60.1%

identity with SicA, all of them were found to bind c-di-GMP with high affinity ($K_d$ = 0.15–0.22 μM) (Supplementary Fig. 22). Thus, our results suggest that the CesD/SycD/LcrH family of T3SS chaperones constitutes a large group of c-di-GMP effectors in Gram-negative pathogenic bacteria. Given the crucial roles of the CesD/SycD/LcrH family of chaperones in the synthesis and/or secretion of T3SS effectors[50–52], our findings support the idea that the activity of T3SSs in a broad range of bacterial pathogens can be modulated by c-di-GMP signaling via this family of chaperones.

## Discussion

As a near-ubiquitous bacterial second messenger, c-di-GMP functions to sense, integrate, and transduce external inputs to allow bacteria to adapt to changing environments, with DGCs and PDEs responsible for its metabolism and effector proteins converting dynamic changes in intracellular c-di-GMP concentration to specific cellular responses[1,4]. However, internal and external cues that directly affect c-di-GMP metabolism as well as c-di-GMP-binding effectors, especially those implicated in bacterial protein secretion and virulence, remain largely unexplored. Here we report that GAPES1 domain-containing DGC YeaJ and its homologs sense the QS signal AI-2 to modulate cellular c-di-GMP levels in a wide range of enteric bacteria, while individual components of bile salts, including taurocholate and taurodeoxycholate, are recognized by the DGC YedQ in S. Typhimurium. Furthermore, we identify the CesD/SycD/LcrH family of T3SS chaperones as c-di-GMP effectors in diverse bacterial pathogens and thus reveal a previously unrecognized and likely widespread mechanism through which dynamic changes in c-di-GMP concentration control bacterial T3SSs.

Although AI-2 is a well-conserved QS signal across the bacterial kingdom[7], it has been associated with relatively few phenotypes in enteric bacteria such as E. coli[53] and S. Typhimurium[54], largely due to the fact that the well-established AI-2 receptor LsrB only mediates AI-2 internalization[55]. Here we resolved this conundrum by identifying a previously unknown AI-2 receptor that possesses DGC activity and harbors an N-terminal GAPES1 domain detecting the AI-2 signal in a diverse range of bacterial species belonging to Enterobacterales, including E. coli, S. Typhimurium and some other enteric bacteria (Supplementary Data 1). This key finding establishes a direct connection between QS and c-di-GMP signaling in these enteric bacteria. Given the global influence of c-di-GMP signaling on bacterial physiology and behavior[1,3,4], we postulate that an array of physiological roles of AI-2 in bacterial species possessing GAPES1 domain-containing AI-2 receptors remain to be explored. As a variety of commensal bacteria inhabiting the mammalian intestinal tract are known to produce AI-2[56], our results also suggest that AI-2 secreted by commensal species might modulate biological processes and cell behavior of invading pathogens like S. Typhimurium and EHEC via the GAPES1-containing AI-2 receptors.

In addition to QS signals, a number of environmental signals have been shown to regulate the activity of CMEs[4–6]. However, whether and how host-derived cues influence bacterial c-di-GMP metabolism remains largely unknown. While bile has been shown to repress the expression of T3SS-1 genes and affect the virulence properties of S. Typhimurium[37,57], our current study links the bile stimulus to changes in c-di-GMP concentration in S. Typhimurium, thus providing a mechanistic explanation for the inhibitory effect of bile on Salmonella invasion of epithelial cells. Furthermore, we show that taurocholate and taurodeoxycholate, two individual components of bile, are responsible for the induction of c-di-GMP through direct binding to the periplasmic sensory domain of YedQ. Perplexingly, we note that the periplasmic LBD (residues 43–357) of YedQ is structurally not well defined. Although the N-terminal region corresponding to residues 41–227 of YedQ matches the CHASE7 domain model (PF17151) at an E-value less than 1E-88 (Supplementary Fig. 23), the putative taurocholate-binding site is located at the C-terminal

region of YedQ-LBD (Fig. 4g–i). Nevertheless, we find that the LBD of the YedQ homolog from EHEC O157:H7 also binds taurocholate and taurodeoxycholate with high affinity (Supplementary Fig. 24), suggesting that YedQ orthologs that respond to bile stimulus might also be widespread among bacteria.

The prominent role of c-di-GMP in bacterial virulence has been well demonstrated in a diverse range of bacterial pathogens[25,58], while several earlier studies have linked c-di-GMP to bacterial secretion systems, including Type II secretion systems (T2SSs), T3SSs, and type VI secretion systems (T6SSs)[18,22,24,59]. A previous study identified the T3SS ATPase HrcN in Pseudomonas syringae as a c-di-GMP-binding effector[59], but the binding of c-di-GMP to HrcN homologs from any other T3SSs has not yet been reported. In addition, we observed no interactions of c-di-GMP with the EscN/YscN/HrcN family of T3SS ATPases from P. aeruginosa (Supplementary Fig. 25a) or S. Typhimurium (Supplementary Fig. 25b), suggesting that binding of c-di-GMP to T3SS ATPases might not be a conserved mechanism controlling bacterial T3SSs. In contrast, the high-affinity binding of c-di-GMP to a group of the CesD/SycD/LcrH family of T3SS chaperones, including those from S. Typhimurium, EHEC, S. flexneri, Y. enterocolitica, P. aeruginosa, and V. parahaemolyticus belonging to Gammaproteobacteria and B. thailandensis within Betaproteobacteria (Fig. 5a and Supplementary Fig. 22), led us to conclude that the regulatory effect of c-di-GMP on T3SSs in a wide range of pathogenic bacteria is mediated by the CesD/SycD/LcrH family of T3SS chaperones.

In conclusion, our work provides insights into not only how QS signals and host-derived cues modulate bacterial c-di-GMP metabolism, but also the molecular mechanism by which c-di-GMP exerts its regulatory effects on bacterial T3SSs (Fig. 6). These findings highlight the complexity of the c-di-GMP signaling networks that may control the same biological processes in response to different extracellular stimuli. As c-di-GMP functions to integrate external inputs from various signaling pathways for decision-making, our study suggests new avenues for the prevention of bacterial infections.

## Methods

### Ethics statement

All animal care and experimental procedures were performed in accordance with the Regulations for the Administration of Affairs Concerning Experimental Animals approved by the State Council of the People's Republic of China. The protocol was approved by the Animal Welfare and Research Ethics Committee of Northwest A&F University (protocol number: NWAFUSM2018001).

### Bacterial strains, plasmid constructions, and growth conditions

All bacterial strains and plasmids used in this study are listed in Supplementary Data 3. All primers used in this study were designed using Primer premier 5.0 (Premier Biosoft) and synthesized by Tsingke Biological Technology (Xi'an, China). Sequences of the primers are listed in Supplementary Data 4. S. Typhimurium strain SL1344 and its derivatives were grown at 30 or 37 °C in either Luria-Bertani (LB) or modified LB medium containing 0.3 M NaCl with or without agitation as indicated. V. harveyi MM32 was grown at 30 °C in AB medium[60]. P. aeruginosa strain PAO1, B. thailandensis E264, V. parahaemolyticus RIMD 2210633, and E. coli strains were cultured in LB medium. In-frame deletion and point mutants of S. Typhimurium were constructed by using the CRISPR-Cas9 system[61]. Genes were cloned into pKT100 for complementation. The DNA fragments encoding 2 SicA homologs and GAPES1 domains of 17 YeaJ homologs were synthesized by Genewiz (Suzhou, China). The E. coli strain BL21(DE3) and its derivative lacking luxS were used for the expression and purification of all His$_6$-tagged recombinant proteins. When required, antibiotics were added to growth media at the following concentrations: ampicillin, 100 μg ml⁻¹; chloramphenicol, 20 μg ml⁻¹; gentamicin, 20 μg ml⁻¹; kanamycin, 50 μg ml⁻¹; streptomycin, 20 μg ml⁻¹.

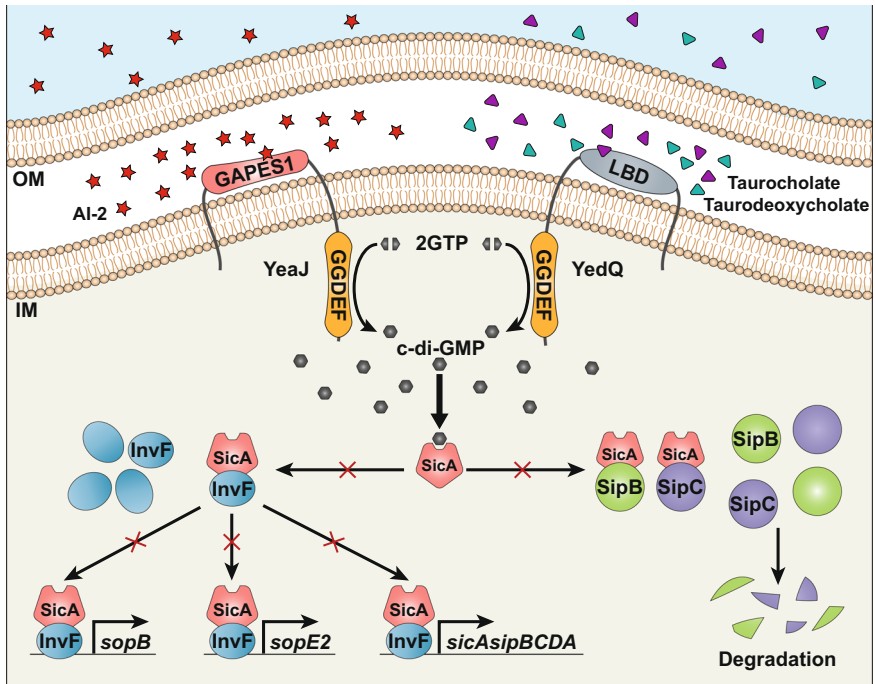

**Fig. 6 | Model of how AI-2 and bile stimulate the synthesis of c-di-GMP to repress the T3SS through targeting the CesD/SycD/LcrH family of chaperones.** The QS signal AI-2 and host-derived cues, including bile components taurocholate and taurodeoxycholate induce an increase in intracellular c-di-GMP concentrations via the DGCs YeaJ and YedQ, respectively. When the intracellular c-di-GMP level is elevated, higher amounts of the T3SS chaperone SicA bound by c-di-GMP will result in less binding of SicA to InvF, SipB, and SipC, thus reducing transcription of the T3SS-1 genes such as *sopB*, *sopE2*, *sicA*, *sipB*, and *sipC* while impairing the pre-secretory stabilization and efficient secretion of SipB and SipC. OM outer membrane, IM inner membrane.

## Biofilm formation and swimming motility assays

Biofilm formation was measured by crystal violet staining[7]. In brief, overnight cultures of *S.* Typhimurium strains were diluted with fresh LB medium to an $OD_{600}$ of 0.05, and 200 µl aliquots of the diluted cells were inoculated into each well of a sterile flat-bottomed 96-well polystyrene microtiter plate. After incubation at 37 °C for 24 h without shaking, culture supernatants were removed, and the wells were washed twice with sterilized water. Cells adhering to the wells were then stained with 0.1% (w/v) crystal violet for 15 min, followed by two washes using distilled water to remove unbound dye. The bacteria-bound dye was redissolved in 200 µl of 95% ethanol and the absorbance was recorded at 590 nm using a microplate reader (Biotek, USA). For swimming motility assay, overnight cultures of *S.* Typhimurium strains were diluted to an $OD_{600}$ of 0.5, and 2 µl aliquots of the diluted cells were spotted onto 0.3% tryptone agar (1% tryptone, 0.5% NaCl, and 0.3% Bacto agar [Difco]). Motility halos were measured after the plates were incubated at 37 °C for 12 h.

## In vitro AI-2 binding assays

*E. coli* strain BL21(DE3) and its mutant lacking *luxS* that carry the pET-28a derivatives containing DNA fragments encoding the LBDs of trans-membrane proteins were cultured initially at 37 °C to an $OD_{600}$ of 0.8, after which isopropyl β-ᴅ-1-thiogalactopyranoside (IPTG) was added to a final concentration of 0.25 mM and then cells were grown for 7 h at 20 °C. After being purified by Ni-nitrilotriacetic acid (Ni²⁺-NTA) His-bind resin (Novagen, Madison, WI) according to the manufacturer's instructions, the His₆-tagged proteins were swapped into a buffer containing 50 mM $NaH_2PO_4$ (pH 8.0), 300 mM NaCl, and 1 mM dithiothreitol using Sephadex-G25 agarose. The purity of purified proteins was assessed by SDS-PAGE and Coomassie brilliant blue staining before protein concentration was measured using the Bradford method. Purified proteins were concentrated to approximately 10 mg ml⁻¹ by using Amicon Ultra centrifugal filters (Millipore) and then heat denatured at 70 °C for 10 min. The denatured proteins were pelleted, and supernatants were subjected to the *V. harveyi* MM32 bioluminescence assay[7]. The AI-2 activity was measured using microplate reader Victor X3 (PerkinElmer, Waltham, MA, USA) run by PerkinElmer 2030 Workstation 4.0.

## ITC analysis

ITC measurements were carried out at 20 °C with a Nano ITC Standard Volume isothermal calorimeter (TA Instruments, New Castle, DE) controlled by the ITCRun software. The N-terminal His₆ tags of all purified proteins were removed before ITC analysis. For titrations with DPD/AI-2 (Omm Scientific), the tag-free proteins were dialyzed against a Tris buffer (25 mM Tris-HCl, 150 mM NaCl, pH 7.5) and diluted to 70 µM, while DPD/AI-2 was diluted with the same buffer to 700 µM. For titrations with 100 µM taurocholate, taurodeoxycholate (both Sigma-Aldrich), SicA, and its variants, the tag-free proteins subjected to the sample cell were dialyzed against the Tris buffer and diluted to 10 µM. Protein samples were dialyzed against the buffer containing 20 mM Tris-HCl (pH 7.5), 100 mM NaCl, and 5 mM $MgCl_2$, and diluted to 10 µM for titrations with 100 µM c-di-GMP, c-di-AMP, or cGMP dissolved in the same buffer. After being degassed, 1 ml of the protein and 250 µl of the ligand solution were added to the sample cell and the syringe, respectively. There were 25 injections per experiment and the stirring speed was 200 rpm. In control experiments, the ligand solution was titrated into the buffer in the sample cell to obtain the heat of dilution. Microcalorimetric data were corrected by subtracting the heats of dilution and fit to the independent binding site model using the NanoAnalyze software.

## Preparation of membrane fractions containing full-length YeaJ and YedQ

The *yeaJ* and *yedQ* genes were cloned with a C-terminal His₆ tag into the vector pHSe5, and the expression constructs were transformed into the Δ*luxS* mutant of *E. coli* BL21(DE3). The resulting strains were grown at 37 °C in LB medium to an $OD_{600}$ of 0.8, and then 0.5 mM IPTG was added to induce protein expression at 20 °C for 10 h. Cells were

harvested by centrifugation and then lysed by sonication in lysis buffer (20 mM Tris-HCl, pH 8.0; 300 mM NaCl; 10% glycerol). Unbroken cells and debris were removed by centrifugation at 10,000×g for 10 min at 4 °C, and then membranes fractions were collected by ultra-centrifugation at 200,000×g for 1 h at 4 °C. The membrane fractions were resuspended in a high-salt buffer (20 mM $Na_3PO_4$, pH 7.0; 2 M KCl; 10% glycerol; 5 mM EDTA; 5 mM DTT; 1 mM phenylmethane-sulfonyl fluoride) and subjected to three rounds (1 h per round) of ultracentrifugation at 350,000×g and 4 °C[62]. The membrane fractions in pellets were resuspended in the high-salt buffer after each round of ultracentrifugation. Membrane fractions were further purified by $Ni^{2+}$-NTA affinity chromatography and subjected to SDS-PAGE to evaluate protein purity (Supplementary Fig. 26). The amount of total proteins in the membrane fractions was quantified by the Bradford assay.

### In vitro DGC activity assays

For YeaJ- and YedQ-mediated production of c-di-GMP, 70 μg of membrane fractions and 100 μM GTP were incubated in a 200-μl reaction system containing 50 mM Tris-HCl (pH 7.5) and 5 mM $MgCl_2$. When needed, 0, 100, or 200 μM DPD/AI-2, taurocholate, or taurodeoxycholate were supplemented. At the time points when the reaction proceeded at 30 °C for 0, 30, and 60 min, 50 μl aliquots were withdrawn from the reaction mixtures and immediately boiled at 100 °C for 5 min. Denatured proteins were removed by centrifugation and then the remaining supernatants were filtered through a 0.22 μm membrane. The levels of c-di-GMP in the supernatants were determined by HPLC analysis[7].

### Cellular nucleotide extractions and c-di-GMP quantification by LC-MS/MS

Relevant strains of S. Typhimurium were cultured at 37 °C in either LB broth with shaking at 200 rpm to $OD_{600}$ of 1.3 or in modified LB medium containing 0.3 M NaCl without agitation to $OD_{600}$ of 1.1. For stimulation with AI-2, the DPD/AI-2 solution was added to the shaking culture at $OD_{600}$ of 1.3, followed by another 30 min of incubation at 37 °C. For induction by porcine bile salts and individual bile components (Sigma-Aldrich), substances were added to the static culture at $OD_{600}$ of 1.1, followed by another 30 min of incubation at 37 °C. About 15 ml of culture was centrifuged and the cells were washed twice with phosphate-buffered saline (PBS, pH 7.0). The cell pellets were resuspended in 500 μl of extraction solution (acetonitrile/methanol/water, 40:40:20, v/v/v) and cooled on ice for 15 min. The samples were boiled for 10 min at 95 °C and then cooled again on ice for 15 min. After centrifugation at 20,000×g and 4 °C for 10 min, supernatants were transferred to new tubes on ice. The remaining pellets were used for two more extraction steps with 500 μl of the extraction solution without heating. Pooled supernatants for each sample were lyophilized, and pellets were resolubilized in 200 μl of distilled water. C-di-GMP concentrations were then measured by LC-MS/MS[63] (AB SCIEX QTRAP 6500+ LC-MS/MS System). Intracellular levels of c-di-GMP were normalized to the number of bacterial cells for each sample.

### Intracellular accumulation and secretion analyses

Overnight cultures of S. Typhimurium strains in LB medium were diluted to an $OD_{600}$ of 0.05 in fresh modified LB medium containing 0.3 M NaCl, and then grown at 37 °C without shaking until the $OD_{600}$ reached 1.1. After centrifugation at 10,000×g for 10 min, the cell pellets and supernatants were separated. To investigate the intracellular accumulation of the target proteins, the cell pellet from 2.5 ml of culture was resuspended in 100 μl of 1× SDS-PAGE loading buffer and then boiled for 10 min at 100 °C. Each 150 ml of culture supernatant was filtered through a 0.22-μm filter (Millipore) to remove the remaining bacteria and other debris. The resultant supernatant was thereafter filtered three times through a nitrocellulose membrane (BA85, Whatman). The membrane retaining secretion proteins was cut into small pieces and resuspended in 100 μl of SDS-PAGE sample loading buffer, followed by 10 min incubation at 100 °C. About 10 μl aliquots of protein samples from both cell pellet and culture supernatant were resolved by SDS-PAGE, followed by western blot analysis using specific antibodies. ICDH was probed as a loading control.

### Western blot analysis

$His_6$-tagged proteins SipB, SipC, and SopB were injected separately into rabbits to generate polyclonal antibodies, and the respective IgG fractions were purified using Protein A Sepharose (GE Healthcare). The antibody specific to ICDH is a kind gift from professor Zhao-Qing Luo at Purdue University[64]. Antibodies against HA-tag and His-tag were purchased from Abways (Shanghai, China). For western blots, protein samples resolved by SDS-PAGE were transferred onto polyvinylidene fluoride membranes (Millipore). After blocking with QuickBlock™ Blocking Buffer (Shanghai Beyotime Biotechnology, China) for 8 h at 4 °C, membranes were incubated overnight at 4 °C with the appropriate primary antibody: rabbit anti-SipB, 1:1000; rabbit anti-SipC, 1:1000; rabbit anti-SopB, 1:1000; rabbit anti-ICDH, 1:5000; mouse anti-His (clone 3B5) (cat# AB0002), 1:5000; mouse anti-HA (clone 3D1) (cat# AB0004), 1:5000. The membranes were then washed five times with TBST buffer (10 mM Tris-HCl, 150 mM NaCl, 0.05% Tween, pH 7.5) and incubated with 1:10,000 dilution of goat anti-rabbit horseradish peroxidase-conjugated secondary antibodies (DIYIBIO, China, cat# DY60202) or goat anti-mouse horseradish peroxidase-conjugated secondary antibodies (DIYIBIO, China, cat# DY60203) at 4 °C for 4 h. After another seven washes with TBST buffer, chemiluminescent signals were detected by using the ECL Plus Kit (GE Healthcare). Uncropped images of blots can be found in the Source Data file.

### RNA extraction and qRT-PCR analysis

S. Typhimurium strains were grown in a modified LB medium containing 0.3 M NaCl without agitation to an $OD_{600}$ of 1.1, and then bacterial cells were collected by centrifugation. Total RNA was extracted using RNAprep Pure Cell/Bacteria Kit (Tiangen Biotech, Beijing, China) and treated with DNase I (Sigma-Aldrich). cDNA was synthesized using TransScript II One-Step gDNA Removal and cDNA Synthesis SuperMix (TransGen Biotech, Beijing, China), and then subjected to qRT-PCR using KAPA SYBR FAST qPCR Kit (Kapa Biosystems, USA) in a LightCycler 96 thermocycler (Roche). To normalize mRNA abundance, the expression of the 16 S rRNA gene was used as an internal control.

### Construction of promoter-lacZ fusion reporter strains and β-galactosidase assays

The promoter regions of sopB, sopE2, and the sicAsipBCDA operon were individually cloned into the vector pDM4-lacZ and the resulting constructs were separately transformed into E. coli strain S17-1λpir. The E. coli S17-1λpir derivatives carrying the promoter-lacZ fusion reporter plasmids were mated with S. Typhimurium strains, and then the recipient S. Typhimurium cells with the respective reporter plasmid integrated into the chromosome were selected on LB plates containing chloramphenicol and streptomycin. The promoter-lacZ fusion reporter strains were grown at 37 °C in a modified LB medium containing 0.3 M NaCl without agitation until their $OD_{600}$ reached 1.1. When needed, porcine bile salts or individual bile components (Sigma-Aldrich) were added to the bacterial cells at $OD_{600}$ of 1.1, followed by incubation for 30 min at 37 °C. The β-galactosidase activity was measured according to the Miller method[65] and was expressed as Miller units.

### Adherence and invasion assays

The human colonic epithelial Caco-2 cell line was purchased from Procell Life Science & Technology Co., Ltd. (cat# CL-0050, Wuhan, China). Caco-2 cells were propagated in Dulbecco's modified Eagle's

medium (DMEM; Gibco) supplemented with 10% fetal bovine serum (FBS; Gibco) and 1% penicillin/streptomycin (Gibco). After expansion, Caco-2 cells were reseeded at a density of $1 \times 10^5$ cells per well in 24-well dishes in DMEM containing 10% FBS and incubated for 48 h at 37 °C in a humidified incubator with 5% $CO_2$ prior to bacterial infections. S. Typhimurium strains were cultured in modified LB medium containing 0.3 M NaCl without agitation to an $OD_{600}$ of 1.1, after which bacterial cells were collected, washed, and resuspended in PBS at a final concentration of $5 \times 10^7$ colony-forming units (CFU) ml$^{-1}$. Caco-2 cells established as monolayer cultures in 24-well plates were infected with S. Typhimurium strains at a multiplicity of infection (MOI) of 50 and incubated for 1 h at 37 °C under a humidified atmosphere with 5% $CO_2$. Infected monolayers were then washed three times with sterile PBS. For the adherence assay, infected monolayers were lysed in 1% Triton X-100, and then serial dilutions were plated on LB agar plates to enumerate bacteria associated with the cell layer. For the invasion assay, the infected cells washed with PBS were further incubated with DMEM supplemented with 10% fetal bovine serum and 100 μg ml$^{-1}$ gentamicin for 2 h at 37 °C in a 5% $CO_2$ incubator. Thereafter, media with gentamicin were removed and monolayers were washed with sterile PBS three times prior to lysis and enumeration of bacteria that had invaded Caco-2 cells by serial dilution and plating. Bacterial adherence was reported as the total number of bacteria associated with the cell layer per well and bacterial invasion was expressed as the ratio of intracellular bacterial counts to the total adherent bacteria.

## Murine infection studies

Six-week-old female BALB/c mice were purchased from Beijing Vital River Laboratory Animal Technology Co., Ltd. (Beijing, China). Four to six mice per cage were housed in a 12-h light:12-h dark cycle at constant temperature (24 ± 2 °C) and humidity (50–60%). After adaption in the lab for 3 days, mice were orally gavaged with streptomycin (10 mg per mouse in 100 μl of saline) 24 h prior to the bacterial challenge. For in vivo competition assay, mice were orally gavaged with a 1:1 mixture of two S. Typhimurium strains in 100 μl of PBS ($5 \times 10^8$ CFU for each strain). On day 2 post-inoculation, feces were collected and then infected mice were sacrificed. The cecum and small intestine are quickly harvested, after which the tissue and feces samples were homogenized, serially diluted, and plated on selective LB agar plates containing 20 μg ml$^{-1}$ streptomycin together with 50 μg ml$^{-1}$ kanamycin or 20 μg ml$^{-1}$ chloramphenicol for CFU enumeration. A competitive index was calculated as the ratio of the test strain to the control strain recovered from mice, divided by the ratio of the two strains in the input[66]. For the survival assay, mice pretreated with streptomycin for 24 h were orally gavaged with $10^7$ CFU of each S. Typhimurium strain. Infected mice were observed three times daily for survival and survival rates were calculated for each strain. Mice without streptomycin treatment were also injected intraperitoneally with $10^4$ CFU of the S. Typhimurium wild-type, mutant, or complemented strains to observe the survival rate.

## Co-IP assays

sicA gene or its derivative with a point mutation (N70A) was cloned with a C-terminal HA-tag into the plasmid pKT100, and then the resulting recombinant plasmids were transformed into S. Typhimurium strain SL1344. The invF, sipB, and sipC genes were cloned with a C-terminal His$_6$ tag into pBBR1MCS1, after which the three derivatives of pBBR1MCS1 were separately transformed into S. Typhimurium SL1344 carrying pKT100-sicA-HA or pKT100-sicA(N70A)-HA. The strains producing both HA-tagged and His$_6$-tagged proteins were cultured in LB medium with shaking at 200 rpm to an $OD_{600}$ of 2.0 at 37 °C. Cells were harvested by centrifugation and then lysed with BugBuster protein extraction reagent (Novagen). Lysates were clarified by centrifugation at $15,000 \times g$ for 10 min at 4 °C and subsequently used for immunoprecipitation experiment with EZview Red Anti-HA

Affinity Gel (Sigma-Aldrich). Different concentrations of nucleotides (c-di-GMP, c-di-AMP, or cGMP) and 20 μl of the pre-washed Anti-HA Affinity Gel were added to 1 ml of lysate and each immunoprecipitation sample was incubated for 1 h at 4 °C. After centrifugation, protein-bead complexes were washed five times with Tris-HCl buffer (pH 7.5) at 4 °C and resuspended in 100 μl of 1× SDS-PAGE loading buffer. Samples were boiled for 10 min and subjected to SDS-PAGE followed by western blot with anti-HA and anti-His antibodies.

## EMSAs

DNA probes were amplified and purified on 6% native polyacrylamide gels. 50 ng of DNA probes, tag-free InvF and SicA, and varying amounts of nucleotides (c-di-GMP, c-di-AMP, or cGMP) were incubated in 20 μl of binding buffer (20 mM Tris-HCl [pH 7.4], 4 mM $MgCl_2$, 100 mM NaCl, 1 mM dithiothreitol, 10% glycerol, and 1% NP-40) for 30 min at room temperature. The samples were then subjected to a 6% native polyacrylamide gel and run in 0.5× Tris-borate-EDTA buffer at 100 V for 3 h at 4 °C. Gels were stained with SYBR Safe DNA gel stain (Invitrogen) and imaged by a fluorescent imaging system (Tanon 5200Multi, China). Uncropped images of gels can be found in the Source Data file.

## Molecular docking analysis

A homology model of SicA was constructed via the Phyre2 server[46] (www.sbg.bio.ic.ac.uk/phyre2) based on the crystal structure of IpgC from S. flexneri (PDB ID: 3GYZ; https://www.rcsb.org/structure/3GYZ). The tertiary structures of YedQ-LBD, InvF, SipB, and SipC were calculated by AlphaFold2 using the multiple sequence alignment (MSA) option[40]. Potential pockets and cavities of the structural models of SicA and YedQ-LBD were calculated using the web-based POCASA 1.1 with a probe radius of 2 Å[47]. AutoDock4[67] was used to add hydrogens and compute Gasteiger charges. The 3D structure of c-di-GMP was extracted from the crystal structure of the c-di-GMP-MapZ complex (PDB ID: 2L74; https://www.rcsb.org/structure/2L74) and the structure of taurocholate was downloaded from the ZINC database (ZINC8214684). Flexible torsions of c-di-GMP and taurocholate were assigned using the Autotors utility within Autodock4. Docking simulations were done with AutoDock Vina 1.1.2[41] and the best binding mode was selected based on the lowest docking energy. Docking simulation of SicA with InvF, SipB, and SipC was performed using the Cluspro 2.0 web server (https://cluspro.org)[48]. The three-dimensional figures were displayed using PyMOL (http://www.pymol.org) and protein–ligand interactions were analyzed using LigPlot+[68].

## Chromosomal promoter replacement in S. Typhimurium

Replacement of the sicAsipBCDA promoter with the invF promoter in S. Typhimurium strains was performed using the CRISPR-Cas9 system[61]. In brief, the invF promoter sequence and the upstream and downstream regions of the sicAsipBCDA promoter amplified by PCR were fused together by overlap extension PCR. The sgRNA fragment that contains a 20-bp guide sequence targeting the sicAsipBCDA promoter was amplified from the plasmid pTargetF1 and was ligated with the fused fragment by overlap extension PCR. The overlapped fragment was cloned into pTargetF1, and then the resulting pTargetF1 derivative was electroporated into S. Typhimurium competent cells harboring the plasmid pCas. S. Typhimurium derivatives with the sicAsipBCDA promoter replaced by the invF promoter were identified by PCR and DNA sequencing, followed by the successive elimination of the pTargetF1 derivative and pCas by IPTG induction and incubation at 37 °C, respectively[61].

## Bioinformatic surveys for YeaJ and SicA homologs and phylogenetic analysis

Searches for YeaJ and SicA homologs were performed using the BLASTP program against the NCBI non-redundant protein database with a cutoff E-value of 1E-03 and a coverage threshold of 60%. For YeaJ

homologs, sequences were further analyzed using InterProScan 5[69] against the Pfam 34.0 database, and only the sequences that possess an N-terminal GAPES1 domain and a C-terminal GGDEF domain with a cutoff E-value of 1E-10 remained for further analysis. The resulting YeaJ and SicA homologs are listed in Supplementary Data 1 and Supplementary Data 2, respectively. For high-confidence YeaJ and SicA homologs that could be unambiguously assigned to genus level, one representative sequence for each genus was randomly selected and subjected to phylogenetic analysis. Multiple protein sequence alignments were performed with clustalX version 1.81 and phylogenetic trees were constructed with MEGA7 software[70] using the maximum-likelihood method based on the Jones-Taylor-Thornton model.

### Statistical analysis

Statistical analysis was performed using Microsoft Excel 2019 or GraphPad Prism 7.0 software. Differences in competitive indexes were analyzed using a two-tailed Mann–Whitney $U$-test. Differences in mouse survival were examined by Log-rank (Mantel–Cox) test. All other experiments were analyzed using the two-tailed unpaired Student's $t$-test. Data were presented as mean ± s.d. or s.e.m. $P$ values less than 0.05 were considered statistically significant.

### Reporting summary

Further information on research design is available in the Nature Portfolio Reporting Summary linked to this article.

## Data availability

The protein 3D coordinate data used in this study are available in the PDB database under accession codes 3GYZ and 2L74. The structure of taurocholate used in this study is available in the ZINC database under accession code ZINC8214684. All the other data that support the findings of this study are available within the paper and its Supplementary Information and Supplementary Data or from the corresponding authors upon request. Source data are provided with this paper.

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

## Acknowledgements

We thank Dr. Sheng Yang at the Institute of Plant Physiology and Ecology, Chinese Academy of Sciences for providing the CRISPR-Cas9 System, and Dr. Zhao-Xun Liang at Nanyang Technological University for providing the plasmid pHSe5. This work was supported by grants from the National Key R&D Program of China 2018YFA0901200 (to L.Z.), and 2021YFC2301403 (to S.O.), the National Natural Science Foundation of

China 32170048 (to L.Z.), 31725003 (to X.S.), 82225028 and 82172287 (to S.O.), and the Special Funds of the Central Government Guiding Local Science and Technology Development 2020L3008 (to S.O.). We thank Dr. Ying Fu (Public Technology Service Center Institute of Microbiology, Chinese Academy of Sciences) for her help in the quantification of intracellular c-di-GMP levels with liquid chromatography-tandem mass spectrometry. We also thank the Teaching and Research Core Facility at the College of Life Science (Min Duan and Ningjuan Fan) and Life Science Research Core Services, NWAFU (Luqi Li) for technical support.

## Author contributions

L.Z. conceived the ideas and designed the experiments; Unless otherwise specified, S.L. performed all of the experiments and conducted data analysis. S.L., Jianghan Li, Y.Z., R.W., C.D., Jialin Li, and Z.W. performed the gene deletions. H.S., L.Z., and S.L. performed the bioinformatic analyses. S.L., Q.L., and R.W. performed the murine infection assays. H.S., L.Z., and S.L. participated in the molecular docking analysis. L.Z., X.S., S.O., Y.W., and L.X. analyzed data and interpreted the results. L.Z., S.L., and X.S. wrote the paper. All authors read and approved the final version of the paper.

## Competing interests

The authors declare no competing interests.
