## [Peer review file · Nature Communications]

REVIEWER COMMENTS

Reviewer #1 (Remarks to the Author):

This paper reports a number of observations related to c-di-GMP signaling, which although interesting, do not coalesce into a cohesive story. The authors report that in *S. Typhimurium* the quorum sensing agonist AI-2 stimulates an increase in c-di-GMP concentration by binding the diguanylate cyclases YeaJ. They also report that the bile salt components taurocholate and taurodeoxycholate also stimulate the production of c-di-GMP by binding another diguanylate cyclases, YeaJ. The authors then attempt to link c-di-GMP signaling to virulence by reporting the potentially interesting observations that c-di-GMP bindings regulates the activity of SicA, a dual-function chaperone of the *S. Typhimurium* type III protein secretion system encoded within its pathogenicity island 1 (SPI-1-TTSS). The authors attempt, but I would argue fail, to connect all these observations into a cohesive story. As a result, the current title for example, which emphasizes arguably the weakest aspect of the paper, does not faithfully represent the main thrust of the study. Although the data are in general of high quality, on several occasions the authors over state their findings drawing conclusions not supported by the data. This is more pertinent to the data linking c-di-GMP to virulence. In general, however, although not cohesively presented, there is some really good stuff in here that would be of interest to those in the field and therefore should find its way into publication. I honestly have little to suggest on how to make the paper more cohesive other than breaking it up into two, which I would not favor. I do think that the authors should at least choose a title that better captures the content of the paper so that readers would be better alerted of its content. I personally do not think that the emphasizes of the title should be in the connection of c-di-GMP to the type III secretion chaperones, which is not only a minor part of the paper but also its weakest. Below, I have some specific comments that the authors should address and that make the paper stronger.

1) The studies on the role of LuxS, YeaJ, and AI-2 in biofilm formation, while well-executed, do not add much to the body of knowledge already available; the results are largely confirmatory or expected. For example, the presence of YeaJ homologs in other enterobacteria has already been reported multiple times (see for example PMID: 29196655). The authors should perhaps consider moving some of these data to supplementary materials to increase the cohesiveness of the paper. They should also indicate when the results are confirmatory of previous studies and when they are new.

2) Previous studies have reported that, contrary to what is stated in this paper, a Δ luxS mutant does not affect biofilm formation (see PMID: 29580499). The incongruity may be related to differences in experimental conditions but the authors should clarify this issue and also indicate when the results are confirmatory of previous studies.

3) The studies to monitor the activity of the activity of the T3SS are not of high quality. For example, no quantitation of the secretion assay shown on Fig. 3a is provided, which given the subtle phenotype observed is essential. The experiment should be conducted several times and should be quantified. Also,

the phenotypes observed are rather minor and yet, the authors used adjectives that are not consistent with the observed phenotype. For example, the authors report a slight increase in the number of intracellular bacteria in the absence of luxS or yeaJ, which they categorize as “dramatic”. To put these numbers in perspective, a lack of the SPI-1 T3SS results in 1,000 fold drop in invasion, which surprisingly does not translate into a dramatic mouse virulence phenotype and only when bacteria are orally administered. In this context the observed phenotype could hardly be characterized as “dramatic”. Also, the results shown contradict previous observations (see for example <https://journals.asm.org/doi/10.1128/JB.00727-09>). The authors must address the incongruity.

4) Something is not right with the competition assays shown in Fig. 3d. Competitions that according to the model presented by the authors are expected to yield similar results showed drastically different indexes. For example, why would Δ yeaJ vs WT would yield an index of 21 while Δ luxS vs WT would give only ~ 2 in some tissues (??). In this type of assays competition indexes of 2 vs 21 are drastically (in this case “drastic” is the appropriate characterization) different. Something is not right with these experiments and may be related to the antibiotic markers. Competitions with the same strains but swapping the markers must be conducted to clarify the issue. Otherwise, it looks like the Δ yeaJ mutant has a much more drastic phenotype than the Δ luxS mutant, which is inconsistent with the proposed model.

5) The authors attempt to link the virulence phenotypes associated with the Δ luxS or Δ yeaJ mutants to differences in the activity of the SPI-1 T3SS but the data do not warrant such connection. The authors cannot rule out the virulence phenotypes observed in this mutant are not due to effects of the mutations on virulence traits other than the T3SS. Specific experiments should be conducted to separate these two issues: 1) compare the virulence of a Δ yeaJ (or Δ luxS) with the phenotype of a Δ yeaJ / Δ TTSS (e. g. combined with a mutant in an essential component of the TTSS). 2) Conduct a comparison of the strains shown in Fig. 3 panel b after intraperitoneal administration, a route in which the T3SS encoded within the pathogenicity island 1 has no virulence defect.

6) An essential experiment that is missing in this study is the examination of the virulence defects as well as T3SS function of a SicA mutants carrying changes in the amino acids that the authors have mapped as essential for c-di-GMP binding. It is surprising that those experiments have not been included in this submission. Without these experiments, along with a demonstration that the mutations do not affect the stability of SicA, the authors cannot draw any definitive conclusions on the role of c-di-GMP on SicA activity.

Reviewer #2 (Remarks to the Author):

This manuscript presents a series of in depth experiments to investigate c-di-GMP signaling in enteric pathogens. A number of novel discoveries are reported regarding YeaJ and YedQ which expands the knowledge around membrane bound receptors and deciphering extra-cytoplasmic signals. The identification and characterization of a GAPES1 domain is also a novel discovery. Perhaps the most intriguing aspect of the manuscript is the role of c-di-GMP binding to SicA (CesD/SycD/LcrH family of T3SS chaperones).

Most of the experimental work is very thorough and the experiments are well controlled. The hypothesis that c-di-GMP binds to SicA is supported by in vitro analyses along with some modelling work. The data here appears to be solid, however the major shortcoming is the absence of an in vivo observation (within bacteria) that would strengthen the hypothesis that c-di-GMP binds to SicA to repress T3SS-1 functionality. Given the breadth and impressive amount of work in parallel experiments, it seems the authors have not directly tested the functional role of c-di-GMP binding to SicA in Salmonella, particularly with respect to functional outcomes.

Recommendations (Major):

1) The data indicates that c-di-GMP interferes with SicA binding to InvF. It apparently interferes with binding to SipB and SipC. Do all these proteins share a common protein interaction surface and is that surface bound by c-di-GMP? Alternatively, is there some sort of allosteric interference? While some modelling data for the SicA interface with c-di-GMP is provided, it is unclear how other interactions would be impacted. This is important in context of the authors model (Fig 7).

2) Specific SicA variants (mutations) that negatively impact on c-di-GMP binding were identified and characterized. Do these mutations impact on InvF binding? SipB? SipC?

3) Given that this would very likely be the first report of a T3SS binding to c-di-GMP, it is essential that the data be strong, exhaustive and supported with complementary and independent experimental assays. As stated above, in vivo evidence for the chaperone interaction model is modest and needs to be provided for the novel claim to be reliably and accurately interpreted. A critical experiment would be to introduce a SicA variant that does not bind c-di-GMP into Salmonella and to demonstrate that T3SS-1 is not responsive to AI-2 or YeaJ absence. Hence a SicA variant (perhaps T25, K27, D28 etc) should be introduced into a yeaJ mutant (hyper-mortality and hyper-invasion). The expected outcome should restore near normal mortality and invasion. The same experiment could be performed with a WT strain, essentially providing evidence that SicA binding to c-di-GMP within the bacteria has a functional outcome that impacts on T3SS-1 activity. This is a critical experiment which needs to be done given the first time claim for T3SS chaperone interaction with c-di-GMP.

Reviewer #3 (Remarks to the Author):

This study identified YeaJ as a GAPES domain-carrying cdG cyclase responsible for sensing the AI-2 autoinducer molecule leading to increased cdG levels in the cell in response. This increase in cdG levels was linked to alterations in the virulence profile of the pathogen through direct interaction with chaperones of the type III secretion system. Another cyclase (YedQ) has also been characterised and found to sense bile components also leading to increased cdG levels in the cell. These results were combined to create a model in which gram negatives such as Salmonella sense triggers in their environment such as bile components or increasing cell density and respond by increasing intracellular cdG levels, which in turn affect phenotypes such as biofilm formation, motility and virulence.

Although conclusions of this work seem to be based on solid experimental data, I found the manuscript very hard to follow.

My main concerns are:

- The absence of an appropriate introduction. I felt that the introduction provides only very superficial and basic information on cdG metabolism, sensing and expression. Pre-existing knowledge is assumed in all results sections of the manuscript. A proper introduction on quorum sensing, type III secretion system effectors and link to cdG would have greatly helped understand the scope of the manuscript.
- The lack of focus. It is clear that the authors have produced an enormous amount of data, which seems to be solid. I feel though that by trying to fit all the results in a single study, it is very hard for the reader to pinpoint the most important messages and understand how all these results are linked together.
- Narration of the manuscript. Result sections are not appropriately separated or labelled. It is not clear how they are linked to each other and how all these results fit together.

In conclusion, although this work seems to be based on solid data; the lack of focus, an appropriate introduction and clear narration render it unsuitable for publication at its current state.

Response to Reviewers

We wish to begin by thanking the three reviewers for their very supportive and constructive comments. Please find our detailed responses to each of the comments below.

Reviewer #1 (Remarks to the Author):

This paper reports a number of observations related to c-di-GMP signaling, which although interesting, do not coalesce into a cohesive story. The authors report that in *S. Typhimurium* the quorum sensing agonist AI-2 stimulates an increase in c-di-GMP concentration by binding the diguanylate cyclases YeaJ. They also report that the bile salt components taurocholate and taurodeoxycholate also stimulate the production of c-di-GMP by binding another diguanylate cyclases, YeaJ. The authors then attempt to link c-di-GMP signaling to virulence by reporting the potentially interesting observations that c-di-GMP bindings regulates the activity of SicA, a dual-function chaperone of the *S. Typhimurium* type III protein secretion system encoded within its pathogenicity island 1 (SPI-1-TTSS). The authors attempt, but I would argue fail, to connect all these observations into a cohesive story. As a result, the current title for example, which emphasizes arguably the weakest aspect of the paper, does not faithfully represent the main thrust of the study. Although the data are in general of high quality, on several occasions the authors over state their findings drawing conclusions not supported by the data. This is more pertinent to the data linking c-di-GMP to virulence. In general, however, although not cohesively presented, there is some really good stuff in here that would be of interest to those in the field and therefore should find its way into publication. I honestly have little to suggest on how to make the paper more cohesive other than breaking it up into two, which I would not favor. I do think that the authors should at least choose a title that better captures the content of the paper so that readers would be better alerted of its content. I personally do not think that the emphasizes of the title should be in the connection of c-di-GMP to the type III secretion chaperones, which is not only a minor part of the paper but also its weakest. Below, I have some specific comments that the authors should address and that make the paper stronger.

Response: We would like to thank the reviewer for the insightful comments

and constructive suggestions. To better capture the content of the paper, we have changed the paper title as “Autoinducer-2 and bile salts induce c-di-GMP synthesis to repress the T3SS via the CesD/SycD/LcrH family of chaperones” We have rearranged the order of the result sections and rewritten the subheadings to make all the observations coalesce into a cohesive story. We have also included new results to support our conclusions and have made the necessary modifications and more detailed explanations to improve the manuscript. In addition, we try our best to avoid overstating our findings.

1) The studies on the role of LuxS, YeaJ, and AI-2 in biofilm formation, while well-executed, do not add much to the body of knowledge already available; the results are largely confirmatory or expected. For example, the presence of YeaJ homologs in other enterobacteria has already been reported multiple times (see for example PMID: 29196655). The authors should perhaps consider moving some of these data to supplementary materials to increase the cohesiveness of the paper. They should also indicate when the results are confirmatory of previous studies and when they are new.

Response: We thank the reviewer for this very insightful point. As suggested, we have moved some of the data that have been reported in previous studies to supplementary materials, including the role of LuxS and YeaJ in biofilm formation and the role of YeaJ in intracellular c-di-GMP levels. When the results are confirmatory of previous studies, they have been indicated along with references to the original literatures in the revised manuscript.

2) Previous studies have reported that, contrary to what is stated in this paper, a $\Delta luxS$ mutant does not affect biofilm formation (see PMID: 29580499). The incongruency may be related to differences in experimental conditions but the authors should clarify this issue and also indicate when the results are confirmatory of previous studies.

Response: We thank the reviewer for this important point. Indeed, a previous study by Ju et al. (PMID: 29580499) suggested that the QS signal AI-2 is not directly related to biofilm formation in *Salmonella* serovar Dublin. However, our study and the study by Ju et al. (PMID: 29580499) employed different *Salmonella* serovars. We note that two previous studies (PMID: 15790567 and

19909098) that also employed *Salmonella enterica* serovar Typhimurium showed that the *luxS* mutant formed less biofilm than the wild type. We have indicated that our results are confirmatory of these two studies in the revised manuscript (**Lines 81-82**).

3) The studies to monitor the activity of the T3SS are not of high quality. For example, no quantitation of the secretion assay shown on Fig. 3a is provided, which given the subtle phenotype observed is essential. The experiment should be conducted several times and should be quantified. Also, the phenotypes observed are rather minor and yet, the authors used adjectives that are not consistent with the observed phenotype. For example, the authors report a slight increase in the number of intracellular bacteria in the absence of *luxS* or *yeaJ*, which they categorize as “dramatic”. To put these numbers in perspective, a lack of the SPI-1 T3SS results in 1,000 fold drop in invasion, which surprisingly does not translate into a dramatic mouse virulence phenotype and only when bacteria are orally administered. In this context the observed phenotype could hardly be characterized as “dramatic”. Also, the results shown contradict previous observations (see for example <https://journals.asm.org/doi/10.1128/JB.00727-09>). The authors must address the incongruency.

Response: We thank the reviewer for these important points. As suggested, we have repeated the intracellular accumulation and secretion assay and replaced Fig. 3a with new high-quality figures. The experiment has been conducted three times and the protein bands of SipB and SopB have been quantified by band densitometry (**Fig. 3a**). We have changed “dramatically increased” as “significantly increased” when we report a increase in the number of intracellular bacteria in the absence of *luxS* or *yeaJ* (**Lines 182-183**). As pointed out by the reviewer, a previous study by Perrett et al. (PMID: 19783624) reported that secretion of T3SS-1 effectors and the ability to invade epithelial cells were not altered in $\Delta luxS$ compared to the wild type. We note that different culture conditions were used for both assays in our study and the study by Perrett et al. (PMID: 19783624). Our study used modified LB medium containing 0.3 M NaCl for static cultivation (a condition for induction of the T3SS-1 encoded on SPI-1) and both assays were performed using cultures

grown to mid-exponential phase, when the AI-2 activity in the culture supernatant of the wild-type strain was maximal (**Supplementary Fig. 3b**), while Perrett et al. (PMID: 19783624) used shaking cultures in normal LB medium with an OD₆₀₀ of 1.0 for T3SS-1 secretion assays and in late log phase for invasion assays. We also found no differences between the wild type and $\Delta luxS$ with respect to their ability to invade epithelial cells in the conditions that Perrett et al. used (**Supplementary Fig. 8**). Thus, the discrepancy observed in LuxS regulation of the T3SS-1 and invasion of epithelial cells can be explained by the use of different culture conditions. We have addressed the incongruity in the revised manuscript (**Lines 184-197**).

4) Something is not right with the competition assays shown in Fig. 3d. Competitions that according to the model presented by the authors are expected to yield similar results showed drastically different indexes. For example, why would $\Delta yeaJ$ vs WT would yield an index of 21 while $\Delta luxS$ vs WT would give only ~2 in some tissues (??). In this type of assays competition indexes of 2 vs 21 are drastically (in this case “drastic” is the appropriate characterization) different. Something is not right with these experiments and may be related to the antibiotic markers. Competitions with the same strains but swapping the markers must be conducted to clarify the issue. Otherwise, it looks like the $\Delta yeaJ$ mutant has a much more drastic phenotype than the $\Delta luxS$ mutant, which is inconsistent with the proposed model.

Response: We thank the reviewer for raising this important question. In the competition assay, $\Delta luxS$ vs WT yielded an index of ~2-3 in the small intestine, cecum and feces, while $\Delta yeaJ$ vs WT yielded an index of ~8 in the small intestine, ~27 in the cecum and ~8 in the feces (**Fig. 3f**). Indeed, competitive indexes between $\Delta luxS$ and the wild type are drastically lower than those between $\Delta yeaJ$ and the wild type, especially in the cecum. As suggested by the reviewer, to exclude the effect of antibiotic markers, we have conducted the competition assay with the same strains but swapping the markers. Nevertheless, similar results were also observed, with $\Delta luxS$ vs WT yielding indexes ~3-4 and $\Delta yeaJ$ vs WT giving an index of ~9 in the small intestine, ~22 in the cecum and ~8 in the feces (**Supplementary Fig. 9**). Thus, our results indicate that the $\Delta yeaJ$ mutant has a much more drastic phenotype than the

$\Delta luxS$ mutant in the competition assay. As the proposed model, YeaJ is a diguanylate cyclase (DGC) while AI-2 is able to stimulate its DGC activity (**Fig. 1e**). However, YeaJ does not completely lose its enzyme activity in the absence of AI-2 (**Fig. 1e**), and intracellular c-di-GMP level in the $\Delta luxS$ mutant (**Fig. 1f**) was higher than that in the $\Delta yeaJ$ mutant (**Supplementary Fig. 4**). Moreover, deletion of *yeaJ* in the $\Delta luxS$ mutant further decreased intracellular c-di-GMP concentration (**Fig. 1g**). Intracellular c-di-GMP levels in these *S. Typhimurium* strains were determined when they were grown to mid-exponential phase, when the AI-2 concentration in the culture supernatant of the wild-type strain was maximal (**Supplementary Fig. 3b**). In other growth phase when the AI-2 concentration in the culture supernatant of the wild-type strain is low, differences in intracellular c-di-GMP levels between the wild type and the $\Delta luxS$ mutant could be much smaller, while intracellular c-di-GMP levels between the wild-type strain and the $\Delta yeaJ$ mutant may still remain drastically different. Thus, It is reasonable that the $\Delta yeaJ$ mutant has a more drastic phenotype than the $\Delta luxS$ mutant in the in vivo competition assay.

5) The authors attempt to link the virulence phenotypes associated with the $\Delta luxS$ or $\Delta yeaJ$ mutants to differences in the activity of the SPI-1 T3SS but the data do not warrant such connection. The authors cannot rule out the virulence phenotypes observed in this mutant are not due to effects of the mutations on virulence traits other than the T3SS. Specific experiments should be conducted to separate these two issues: 1) compare the virulence of a $\Delta yeaJ$ (or $\Delta luxS$) with the phenotype of a $\Delta yeaJ / \Delta TTSS$ (e. g. combined with a mutant in an essential component of the TTSS). 2) Conduct a comparison of the strains shown in Fig. 3 panel b after intraperitoneal administration, a route in which the T3SS encoded within the pathogenicity island 1 has no virulence defect.

Response: We thank the reviewer for this insightful suggestion. As suggested, we have compare the virulence of $\Delta luxS$ and $\Delta yeaJ$ with the phenotype of $\Delta invC$ (the gene encoding the T3SS-1 ATPase InvC), $\Delta invC \Delta luxS$ and $\Delta invC \Delta yeaJ$ in mice (**Fig. 3g**). While $\Delta luxS$ and $\Delta yeaJ$ showed enhanced virulence compared with the wild-type strain after oral infection of mice, $\Delta invC$, $\Delta invC \Delta luxS$ and $\Delta invC \Delta yeaJ$ showed reduced virulence compared with the

wild-type strain (**Fig. 3g**). Moreover, $\Delta invC\Delta luxS$ and $\Delta invC\Delta yeaJ$ showed similar virulence compared to $\Delta invC$ (**Fig. 3g**), indicating that AI-2-mediated c-di-GMP signaling regulates the virulence of *S. Typhimurium* via T3SS-1. We have also conducted a comparison of *S. Typhimurium* wild-type, $\Delta luxS$, $\Delta yeaJ$ and complemented strains after intraperitoneal administration, and found that $\Delta luxS$ and $\Delta yeaJ$ showed similar virulence compared to the wild-type strain after intraperitoneal infection (**Supplementary Fig. 10**), suggesting that YeaJ-dependent AI-2-induced repression of the T3SS-1 has no major impact on systemic infection.

6) An essential experiment that is missing in this study is the examination of the virulence defects as well as T3SS function of a SicA mutants carrying changes in the amino acids that the authors have mapped as essential for c-di-GMP binding. It is surprising the those experiments have not been included in this submission. Without these experiments, along with a demonstrations that the mutations do not affect the stability of SicA, the authors cannot draw any definitive conclusions on the role of c-di-GMP on SicA activity.

Response: We thank the reviewer for this insightful comment. In the revised manuscript, we have found that changing N70 to alanine of SicA specifically impaired binding of c-di-GMP (a 140-fold increase in K_d ; **Fig. 5a and Supplementary Fig. 17e**) but did not affect its binding affinities for InvF, SipB and SipC (**Supplementary Fig. 19**). Furthermore, high concentrations of c-di-GMP failed to impair co-immunoprecipitation of InvF-His₆, SipB-His₆ and SipC-His₆ with SicA_{N70A}-HA (**Supplementary Fig. 20**). We thus constructed the point mutant *sicA(N70A)*. While the expression of *sipB*, *sopB* and *sopE2* was significantly reduced in the $\Delta sicA$ mutant compared to the wild-type strain, mRNA levels of these genes were drastically increased in the point mutant *sicA(N70A)* compared to the wild-type strain (**Fig. 5I**). Moreover, deletion of *luxS* or *yeaJ* in the *sicA(N70A)* mutant background did not alter expression of these T3SS-1 genes (**Fig. 5I**). In line with this, mice infected with the $\Delta sicA$ mutant showed significantly lower mortality than those infected with the wild-type strain (**Fig. 5m**). In contrast, mice infected with the *sicA(N70A)* mutant showed significantly higher mortality than those infected with the

wild-type strain, whereas infections with *sicA*(N70A) and its derivative mutants lacking *luxS* or *yeaJ* produced similar mortality (**Fig. 5m**). These in vivo observations further confirm that c-di-GMP exerts its negative regulatory effects on T3SS-1 through binding to SicA.

Reviewer #2 (Remarks to the Author):

This manuscript presents a series of in depth experiments to investigate c-di-GMP signaling in enteric pathogens. A number of novel discoveries are reported regarding YeaJ and YedQ which expands the knowledge around membrane bound receptors and deciphering extra-cytoplasmic signals. The identification and characterization of a GAPES1 domain is also a novel discovery. Perhaps the most intriguing aspect of the manuscript is the role of c-di-GMP binding to SicA (CesD/SycD/LcrH family of T3SS chaperones).

Response: The authors are very grateful for the positive feedback provided by the reviewer.

Most of the experimental work is very thorough and the experiments are well controlled. The hypothesis that c-di-GMP binds to SicA is supported by in vitro analyses along with some modelling work. The data here appears to be solid, however the major shortcoming is the absence of an in vivo observation (within bacteria) that would strengthen the hypothesis that c-di-GMP binds to SicA to repress T3SS-1 functionality. Given the breadth and impressive amount of work in parallel experiments, it seems the authors have not directly tested the functional role of c-di-GMP binding to SicA in *Salmonella*, particularly with respect to functional outcomes.

Response: We thank the reviewer for raising this important question. In the revised manuscript, we have constructed the point mutant *sicA*(N70A) in *S. Typhimurium*. We have found that changing N70 to alanine of SicA specifically impaired binding of c-di-GMP (a 140-fold increase in K_d ; **Fig. 5a and Supplementary Fig. 17e**) but did not affect its binding affinities for InvF, SipB and SipC (**Supplementary Fig. 19**). Furthermore, high concentrations of c-di-GMP failed to impair co-immunoprecipitation of InvF-His₆, SipB-His₆ and SipC-His₆ with SicA_{N70A}-HA (**Supplementary Fig. 20**). While the expression of *sipB*, *sopB* and *sopE2* was significantly reduced in the Δ *sicA* mutant compared

to the wild-type strain, mRNA levels of these genes were drastically increased in the point mutant *sicA(N70A)* compared to the wild-type strain (**Fig. 5l**). Moreover, deletion of *luxS* or *yeaJ* in the *sicA(N70A)* mutant background did not alter expression of these T3SS-1 genes (**Fig. 5l**). In an oral infection model, mice infected with the Δ *sicA* mutant showed significantly lower mortality than those infected with the wild-type strain (**Fig. 5m**). In contrast, mice infected with the *sicA(N70A)* mutant showed significantly higher mortality than those infected with the wild type, whereas infections with *sicA(N70A)* and its derivative mutants lacking *luxS* or *yeaJ* produced similar mortality (**Fig. 5m**). These in vivo observations further confirm that c-di-GMP exerts its regulatory effects on T3SS-1 through binding to SicA.

Recommendations (Major):

1) The data indicates that c-di-GMP interferes with SicA binding to InvF. It apparently interferes with binding to SipB and SipC. Do all these proteins share a common protein interaction surface and is that surface bound by c-di-GMP? Alternatively, is there some sort of allosteric interference? While some modelling data for the SicA interface with c-di-GMP is provided, it is unclear how other interactions would be impacted. This is important in context of the authors model (Fig 7).

Response: We thank the reviewer for this very insightful point. In order to answer this question, we have performed protein-protein docking analysis using Cluspro 2.0, which suggests that InvF, SipB and SipC have partially overlapping interaction surfaces on SicA (R61 of SicA participates in interactions with InvF, SipB and SipC) (**Supplementary Fig. 18**). The docking analysis also suggested that the interaction surfaces of SicA with SipB and SipC, but not with InvF, partially overlap with the c-di-GMP-binding site (**Supplementary Fig. 18 and Fig. 5j**). Among residues of SicA that make contact to c-di-GMP, K27, D28, Q34 and D67, but not N70, were predicted to participate in interactions with its protein partners SipB (interacting with K27, D28, Q34 of SicA) and SipC (interacting with D28 and D67 of SicA) (**Supplementary Fig. 18**). Indeed, the K27A variant of SicA showed a 19-fold lower binding affinity to SipB compared with wild-type SicA (**Supplementary Fig. 19**). In contrast, the N70A mutation of SicA did not affect its binding

affinities for InvF, SipB and SipC (**Supplementary Fig. 19**). While the interaction surface of SicA with InvF does not overlap with the c-di-GMP-binding site (**Supplementary Fig. 18**), c-di-GMP interferes with SicA binding to InvF (**Fig. 5b**), suggesting that c-di-GMP binding may exert an allosteric effect on SicA. The inhibitory effect of c-di-GMP on the binding of SicA to SipB and SipC may depend not only on allosteric interference but also the partial overlap of protein interaction surface with the c-di-GMP-binding site.

2) Specific SicA variants (mutations) that negatively impact on c-di-GMP binding were identified and characterized. Do these mutations impact on InvF binding? SipB? SipC?

Response: We thank the reviewer for this important point. As mentioned above, protein-protein docking analysis by Cluspro 2.0 suggested that the interaction surfaces of SicA with SipB and SipC, but not with InvF, partially overlap with the c-di-GMP-binding site (**Supplementary Fig. 18 and Fig. 5j**). Among residues of SicA that make contact to c-di-GMP, K27, D28, Q34 and D67, but not N70, were predicted to participate in interactions with its protein partners SipB and SipC (**Supplementary Fig. 18**). Indeed, the K27A variant of SicA showed a 19-fold lower binding affinity to SipB compared with wild-type SicA, while the N70A mutation of SicA did not affect its binding affinities for InvF, SipB and SipC (**Supplementary Fig. 19**). Moreover, changing N70 to alanine did not affect the ability of SicA to co-immunoprecipitate InvF-His₆, SipB-His₆ and SipC-His₆, whereas high concentrations of c-di-GMP failed to impair co-IP of InvF-His₆, SipB-His₆ and SipC-His₆ with SicA_{N70A}-HA (**Supplementary Fig. 20**). These results indicate that changing N70 to alanine specifically impairs binding of c-di-GMP but leaves the chaperone function of SicA unaffected.

3) Given that this would very likely be the first report of a T3SS binding to c-di-GMP, it is essential that the data be strong, exhaustive and supported with complementary and independent experimental assays. As stated above, in vivo evidence for the chaperone interaction model is modest and needs to be provided for the novel claim to be reliably and accurately interpreted. A critical experiment would be to introduce a SicA variant that does not bind c-di-GMP

into Salmonella and to demonstrate that T3SS-1 is not responsive to AI-2 or YeaJ absence. Hence a SicA variant (perhaps T25, K27, D28 etc) should be introduced into a *yeaJ* mutant (hyper-mortality and hyper-invasion). The expected outcome should restore near normal mortality and invasion. The same experiment could be performed with a WT strain, essentially providing evidence that SicA binding to c-di-GMP within the bacteria has a functional outcome that impacts on T3SS-1 activity. This is a critical experiment which needs to be done given the first time claim for T3SS chaperone interaction with c-di-GMP.

Response: We thank the reviewer for this insightful comment. As suggested, we have found that changing N70 to alanine in SicA drastically impairs its binding with c-di-GMP (a 140-fold increase in K_d ; **Fig. 5a and Supplementary Fig. 17e**) but leaves its chaperone function of SicA unaffected (**Supplementary Fig. 19 and Supplementary Fig. 20**). We thus constructed the point mutant *sicA(N70A)* in *S. Typhimurium*. While the expression of *sipB*, *sopB* and *sopE2* was significantly reduced in the Δ *sicA* mutant compared to the wild-type strain, mRNA levels of these T3SS-1 effector genes were drastically increased in the point mutant *sicA(N70A)* compared to the wild-type strain (**Fig. 5l**). Moreover, deletion of *luxS* or *yeaJ* in the *sicA(N70A)* mutant did not alter expression of these T3SS-1 genes (**Fig. 5l**), demonstrating that T3SS-1 is not responsive to the absence of AI-2 or YeaJ in the *sicA(N70A)* background. In line with this, mice infected with the Δ *sicA* mutant showed significantly lower mortality than those infected with the wild-type strain (**Fig. 5m**). In contrast, mice infected with the *sicA(N70A)* mutant showed significantly higher mortality than those infected with the wild-type strain, whereas infections with *sicA(N70A)* and its derivative mutants lacking *luxS* or *yeaJ* produced similar mortality (**Fig. 5m**). These in vivo evidences further confirm that c-di-GMP exerts its negative regulatory effects on T3SS-1 through binding to SicA.

Reviewer #3 (Remarks to the Author):

This study identified YeaJ as a GAPES domain-carrying cdG cyclase responsible for sensing the AI-2 autoinducer molecule leading to increased cdG levels in the cell in response. This increase in cdG levels was linked to

alterations in the virulence profile of the pathogen through direct interaction with chaperones of the type III secretion system. Another cyclase (YedQ) has also been characterised and found to sense bile components also leading to increased cdG levels in the cell. These results were combined to create a model in which gram negatives such as Salmonella sense triggers in their environment such as bile components or increasing cell density and respond by increasing intracellular cdG levels, which in turn affect phenotypes such as biofilm formation, motility and virulence. Although conclusions of this work seem to be based on solid experimental data, I found the manuscript very hard to follow.

Response: The authors are very grateful for the positive feedback provided by the reviewer. We also thank the reviewer for the insightful comment on the readability of the manuscript. We have reorganized the language and structure to improve the logicity and readability of the manuscript.

My main concerns are:

- The absence of an appropriate introduction. I felt that the introduction provides only very superficial and basic information on cdG metabolism, sensing and expression. Pre-existing knowledge is assumed in all results sections of the manuscript. A proper introduction on quorum sensing, type III secretion system effectors and link to cdG would have greatly helped understand the scope of the manuscript.

Response: We thank the reviewer for raising this important question. As suggested, we have rewritten the introduction, including a proper description of quorum sensing, type III secretion system effectors and their link to the c-di-GMP signaling network.

- The lack of focus. It is clear that the authors have produced an enormous amount of data, which seems to be solid. I feel though that by trying to fit all the results in a single study, it is very hard for the reader to pinpoint the most important messages and understand how all these results are linked together.

Response: We thank the reviewer for this insightful comment. In order to better capture the content of the paper, we have changed the paper title as "Autoinducer-2 and bile salts induce c-di-GMP synthesis to repress the T3SS

via the CesD/SycD/LcrH family of chaperones". We have reorganized the result sections of the manuscript in accordance with the new title so that readers will be better alerted of its content.

- Narration of the manuscript. Result sections are not appropriately separated or labelled. It is not clear how they are linked to each other and how all these results fit together.

Response: We thank the reviewer for the constructive suggestions. In the revised manuscript, we have rearranged the order of the result sections and rewritten the subheadings to make them better linked to each other and coalesce into a cohesive story.

REVIEWER COMMENTS

Reviewer #2 (Remarks to the Author):

The revised manuscript makes considerable improvements to the original version. The main advance is the generation of a SicA variant that does not bind to c-di-GMP yet retains binding to InvF, SipB and SipC.

Unfortunately, the main issue relating to the functional and important regulatory effect of SicA binding to c-di-GMP remains somewhat contentious. The data (Fig 5m) does not align with the author's model. Specifically, SicA that does not bind c-di-GMP should support near WT infection, but the data suggests otherwise. The authors statements around this topic in the rebuttal letter and in the main text were also very confusing and somewhat misleading, at least to this reviewer. The experiment in Fig 5m would greatly benefit from some side-by-side controls for single mutant strains, in addition to genetic complementation. Such data are very much needed for accurate functional relevance of the SicA interaction with c-di-GMP.

The in vitro data for SicA binding to c-di-GMP seems reasonable. I remain less convinced and uncertain about the physiological (functional) relevance of SicA binding to c-di-GMP. There might be other factors at play including pleiotropic genetic effects. If that is the case, the authors should at least consider and present that scenario to address the observations within their datasets.

Response to Reviewers

We wish to begin by thanking the reviewer for the very supportive and constructive comments. In the revised manuscript, we have addressed all concerns from the reviewer to our best and hope our revision and explanations could answer the reviewer's questions. Please find our detailed responses to each of the comments below.

Reviewer #2 (Remarks to the Author):

The revised manuscript makes considerable improvements to the original version. The main advance is the generation of a SicA variant that does not bind to c-di-GMP yet retains binding to InvF, SipB and SipC.

Response: The authors are very grateful for the positive feedback provided by the reviewer.

Unfortunately, the main issue relating to the functional and important regulatory effect of SicA binding to c-di-GMP remains somewhat contentious. The data (Fig 5m) does not align with the author's model. Specifically, SicA that does not bind c-di-GMP should support near WT infection, but the data suggests otherwise. The authors statements around this topic in the rebuttal letter and in the main text were also very confusing and somewhat misleading, at least to this reviewer. The experiment in Fig 5m would greatly benefit from some side-by-side controls for single mutant strains, in addition to genetic complementation. Such data are very much needed for accurate functional relevance of the SicA interaction with c-di-GMP.

Response: We would like to thank the reviewer for the insightful comments and constructive suggestions. We first apologize for the confusing and possible misleading statements relating to the functional and regulatory effect of SicA binding to c-di-GMP. We have made the necessary modifications and more detailed explanations in this content to make the readers better understand them (**Lines 363-392, 1164-1167**). When SicA_{N70A} and wild-type

SicA are expressed at a similar level and separately incubated with an equal amount of InvF and the same concentrations of c-di-GMP (30 and 60 μ M), the amounts of InvF, SipB and SipC bound by SicA_{N70A} were much higher than those bound by wild-type (WT) SicA (Supplementary Fig. 20), which can be attributed to the much lower c-di-GMP-binding affinity of SicA_{N70A} and thus much less binding of SicA_{N70A} to c-di-GMP when compared with WT SicA. Thus, in the WT strain and the *sicA*(N70A) mutant expressing an equal amount of InvF and producing a similar level of c-di-GMP, when the protein level of SicA_{N70A} in the mutant is equivalent to that of WT SicA in the WT strain, the amount of InvF bound by SicA_{N70A} in the *sicA*(N70A) mutant could be much more than the amount of InvF bound by WT SicA in the WT strain (see the model shown below), thus leading to enhanced transcription of the target genes of InvF/SicA such as *sicA/sicA*(N70A), *sipB*, *sopB* and *sopE2* in the *sicA*(N70A) mutant compared with the WT strain (**Fig. 5l**). The higher expression of *sicA*(N70A) in the *sicA*(N70A) mutant will further increase transcription of the target genes of InvF/SicA_{N70A} including *sicA*(N70A), *sipB*, *sopB* and *sopE2*. As a result, the expression levels of the T3SS-1 genes are significantly higher in the *sicA*(N70A) mutant compared to the WT strain (Fig. 5l). **As suggested, we have repeated the mice survival rate assay, with *S. Typhimurium* single mutant strains including Δ *sicA*, Δ *luxS* and Δ *yeaJ* for side-by-side controls.** Complemented strains were also included in the mouse infection assay. The mutants Δ *luxS* and Δ *yeaJ* led to significantly increased mouse mortality compared to the WT strain, while mice infected with the Δ *sicA* mutant showed significantly lower mortality than those infected with the WT strain (**Fig. 5m**). Complementation with the respective wild-type genes returned the lethality of these mutants in mice to WT levels (**Fig. 5m**). Nevertheless, mice infected with the *sicA*(N70A) mutant showed significantly higher mortality than those infected with the WT strain (Fig. 5m), which can be explained by upregulated expression of the T3SS-1 genes and thus enhanced T3SS-1 activity in the *sicA*(N70A) mutant compared to the WT strain (Fig. 5l).

Moreover, infections with *sicA(N70A)*, its derivative mutants lacking *luxS* or *yeaJ* and the corresponding *luxS/yeaJ* complemented strains produced similar mortality (**Fig. 5m**), which can be attributed to similar expression levels of the T3SS-1 genes in these *sicA(N70A)* derivatives (**Fig. 5l**). Thus, our results indicate that the *sicA(N70A)* mutant that expresses SicA_{N70A} with very low c-di-GMP-binding affinity exhibits higher virulence than the WT strain after oral infection of mice (**Fig. 5m**), which is consistent with the differences in T3SS-1 gene expression between the *sicA(N70A)* mutant and the WT strain (**Fig. 5l**). These data are consistent with our proposed model.

Lower amounts of SicA_{N70A} bound by c-di-GMP result in more binding of SicA_{N70A} to InvF, thus enhancing transcription of the T3SS-1 genes

The in vitro data for SicA binding to c-di-GMP seems reasonable. I remain less convinced and uncertain about the physiological (functional) relevance of SicA binding to c-di-GMP. There might be other factors at play including pleiotropic genetic effects. If that is the case, the authors should at least consider and present that scenario to address the observations within their datasets.

Response: We thank the reviewer for the helpful comments and suggestions. We will give some explanations below, and hope our explanations can answer

the doubts of this reviewer. While playing important functions, bacterial secretion systems such as T3SSs, T4SSs and T6SSs require energy transduction systems to power the export of their substrates through the outer membrane. To avoid unnecessary energy consumption, bacteria usually turn these protein secretion machines on when needed and turn down them when their activities are not required. Bacterial secretion systems are regulated precisely by a variety of regulatory systems at transcriptional, translational, and post-translational levels (Leung *et al.* Curr. Opin. Microbiol. 2011, **14**:9-15; Volk *et al.*, Curr. Top. Microbiol. Immunol. 2019, **427**:11-33.), which enables bacteria to adapt to varied environments. Remarkably, c-di-GMP signalling has been shown to regulate T3SSs, T4SSs and T6SSs in pathogenic bacteria such as *Pseudomonas aeruginosa*, *S. Typhimurium* and *Agrobacterium tumefaciens* (Moscoso *et al.* Environ. Microbiol. 2011, **13**:3128-3138; Lamprokostopoulou *et al.* Environ. Microbiol. 2010, **12**:40-53; McCarthy *et al.* Mol. Microbiol. 2019, **112**:632-648.). For example, c-di-GMP is reported to impact T4SS and T6SS of *A. tumefaciens* at the transcriptional level (McCarthy *et al.* Mol. Microbiol. 2019, 112:632-648.). However, the exact role of c-di-GMP in these secretion systems remains unclear. In our study, we also found that c-di-GMP regulates the T3SS-1 of *S. Typhimurium* at the transcriptional level and identified the T3SS-1 chaperone SicA as a target protein of c-di-GMP. When the intracellular c-di-GMP level is elevated, higher amounts of SicA bound by c-di-GMP will result in less binding of SicA to InvF, thus reducing transcription of the T3SS-1 genes such as *sicA*, *sipB*, *sopB* and *sopE2*. It should be noted that *sicA* expression is autoregulated by the InvF/SicA complex. Thus, reduced expression of *sicA* will then lead to further downregulated expression of the T3SS-1 genes including *sicA*, *sipB*, *sopB* and *sopE2*. Through this positive autoregulatory loop, the effects of elevation of intracellular c-di-GMP levels on SicA activity would be amplified significantly in vivo, thus inducing rapid downregulation of the T3SS-1 gene expression. On the other hand, elevated c-di-GMP also leads to less binding of SicA to SipB

and SipC, thus impairing the stability and secretion of SipB and SipC. Thus, the T3SS chaperone SicA could be an efficient c-di-GMP-responsive switch that rapidly modulates the T3SS activity in response to changes in intracellular c-di-GMP concentration. While our results are consistent with the proposed model, we have made the necessary modifications and more detailed explanations to make the readers better understand the content relating to the role of c-di-GMP on SicA activity in vivo (**Lines 363-392, 1164-1167**).

REVIEWERS' COMMENTS

Reviewer #4 (Remarks to the Author):

In the article titled "Autoinducer-2 and bile salts induce c-di-GMP synthesis to repress the T3SS via the CesD/SycD/LcrH family of chaperones", the authors make a series of observations regarding c-di-GMP metabolisms and its relationship to the transcription of the SPI-1 encoded type III secretion system of Salmonella. The observations are interesting, but the authors fail to unite the different observations into a cohesive story.

Regarding the issues raised by "Reviewer #2", I think the authors successfully address most of his/her concerns, and therefore, although I think that this manuscript is really two stories marginally related put together into one (I am not convinced that the authors have demonstrated that the signals transduced by YeaJ and the effects of bile components on YedQ are directly connected to SicA), I think that there is important information for the field, throughout the manuscript that result in a significant contribution and therefore I believe that the paper should be published.

Reviewer #5 (Remarks to the Author):

The manuscript has been revised substantially to include data supporting the increased virulence of the SicA(N70A) mutant relative to the wild-type strain in a mouse model of infection (Fig. 5m). Specifically, the authors show that in comparison to wild-type, the SicA(N70A) mutant binds significantly greater amounts of InvF, SipB, and SipC in the presence of c-di-GMP, and that enhanced binding to InvF leads to increased expression of SPI-1 associated genes (Supplementary Fig. 20, Fig. 5l). Additional mutants and complemented strains were also included in Fig. 5m, and the survival data is consistent with the in vitro work for the SicA(N70A) mutant. Based on these revisions, the concerns raised by Reviewer #2 have been sufficiently addressed (I would like the authors to note that I was not the original Reviewer 2 and was not involved in the original review).

There are, however, some concerns with the inconsistency of how the results for the SicA(N70A) mutant were framed in the text. Lines 346-352 describe that residue N70 is involved in the binding of c-di-GMP, and this site does not overlap with SicA binding to its partner proteins. These data suggest that c-di-GMP allosterically regulates SicA and this is abrogated by an N70A mutation. The text should be clarified as it is misleading as written to indicate that increased binding of SicA(N70A) to SPI-1 proteins is (simply)

attributable to lower levels of c-di-GMP binding (Lines 366-367 and 372) or that SicA(N70A) promotes T3SS-1 activity (Lines 386-387).

Other: It would have been good to include a yedQ mutant in the section on SicA starting at Line 285 to help tie the narrative together on the two diguanylate cyclases and the regulation of SPI-1. Without this, I found that reading about yedQ was a little distracting mid-way through the paper. The parts on the SicA(N70A) mutant could also benefit from some structure(s) to better understand how c-di-GMP regulates SicA, but this is probably beyond the scope of this work.

Response to Reviewers

We wish to begin by thanking the two reviewers for the very supportive and constructive comments. In the revised manuscript, we have addressed the concerns from the reviewers to our best and hope our revision and explanations could answer the reviewer's questions. Please find our detailed responses to each of the comments below.

Reviewer #4 (Remarks to the Author):

In the article titled "Autoinducer-2 and bile salts induce c-di-GMP synthesis to repress the T3SS via the CesD/SycD/LcrH family of chaperones", the authors make a series of observations regarding c-di-GMP metabolisms and its relationship to the transcription of the SPI-1 encoded type III secretion system of Salmonella. The observations are interesting, but the authors fail to unite the different observations into a cohesive story.

Response: The authors are very grateful for the feedback provided by the reviewer. In fact, the manuscript consists of two parts. At the first part, we revealed that autoinducer-2 and bile salts induce c-di-GMP synthesis to repress the T3SS. At the second part, we identified a novel c-di-GMP effector involved in T3SS regulation and thus revealed the mechanism through which increasing intracellular c-di-GMP levels repress the T3SS. We have included the mutants involved in c-di-GMP synthesis in the section on SicA, and try our best to connect the two parts into a cohesive story. Under the guidance of the article title, readers would be better alerted of its content.

Regarding the issues raised by "Reviewer #2", I think the authors successfully address most of his/her concerns, and therefore, although I think that this

manuscript is really two stories marginally related put together into one (I am not convinced that the authors have demonstrated that the signals transduced by YeaJ and the effects of bile components on YedQ are directly connected to SicA), I think that there is important information for the field, throughout the manuscript that result in a significant contribution and therefore I believe that the paper should be published.

Response: We would like to thank the reviewer for the very supportive comments on our study. It is well established that c-di-GMP acts as an intracellular second messenger transducing extracellular stimuli into intracellular signaling events. As mentioned in the manuscript, c-di-GMP functions to sense, integrate, and transduce external inputs to allow bacteria to adapt to changing environments, with DGCs and PDEs responsible for its metabolism and c-di-GMP effector proteins converting dynamic changes in intracellular c-di-GMP concentration to specific cellular responses (**Lines 431-434**). External signals such as AI-2 and bile salts stimulate the DGC activity of YeaJ and YedQ, respectively, to increase intracellular c-di-GMP levels, while c-di-GMP binds its effector SicA to repress T3SS-1 gene expression.

Reviewer #5 (Remarks to the Author):

The manuscript has been revised substantially to include data supporting the increased virulence of the SicA(N70A) mutant relative to the wild-type strain in a mouse model of infection (Fig. 5m). Specifically, the authors show that in comparison to wild-type, the SicA(N70A) mutant binds significantly greater amounts of InvF, SipB, and SipC in the presence of c-di-GMP, and that enhanced binding to InvF leads to increased expression of SPI-1 associated genes (Supplementary Fig. 20, Fig. 5l). Additional mutants and complemented

strains were also included in Fig. 5m, and the survival data is consistent with the in vitro work for the SicA(N70A) mutant. Based on these revisions, the concerns raised by Reviewer #2 have been sufficiently addressed (I would like the authors to note that I was not the original Reviewer 2 and was not involved in the original review).

Response: The authors are very grateful for the positive feedback provided by the reviewer.

There are, however, some concerns with the inconsistency of how the results for the SicA(N70A) mutant were framed in the text. Lines 346-352 describe that residue N70 is involved in the binding of c-di-GMP, and this site does not overlap with SicA binding to its partner proteins. These data suggest that c-di-GMP allosterically regulates SicA and this is abrogated by an N70A mutation. The text should be clarified as it is misleading as written to indicate that increased binding of SicA(N70A) to SPI-1 proteins is (simply) attributable to lower levels of c-di-GMP binding (Lines 366-367 and 372) or that SicA(N70A) promotes T3SS-1 activity (Lines 386-387).

Response: We thank the reviewer for this insightful comment. Although the interaction surfaces of SicA with SipB and SipC partially overlap with the c-di-GMP-binding site, the interaction surface of SicA with InvF does not overlap with the c-di-GMP-binding site (**Supplementary Fig. 18** and **Fig. 5j**). The binding of InvF to SicA was also disturbed by c-di-GMP binding (**Fig. 5b**), suggesting that c-di-GMP allosterically regulates SicA. Furthermore, high concentrations of c-di-GMP failed to impair co-immunoprecipitation of InvF-His₆ with SicA_{N70A}-HA (**Supplementary Fig. 20**), suggesting that allosteric regulation of SicA by c-di-GMP is abrogated by an N70A mutation. The inhibitory effect of c-di-GMP on the binding of SicA to SipB and SipC may

depend not only on allosteric interference but also the partial overlap of protein interaction surface with the c-di-GMP-binding site. In fact, the inability of c-di-GMP to bind SicA_{N70A} (the very low c-di-GMP-binding affinity of SicA_{N70A} has been demonstrated in **Fig. 5k**) can explain why c-di-GMP failed to allosterically regulate SicA_{N70A}. The speculations about the allosteric regulation of SicA by c-di-GMP and its abrogation by an N70A mutation have been supplemented in the revised manuscript (**Lines 363-365, 374-375**).

Other: It would have been good to include a yedQ mutant in the section on SicA starting at Line 285 to help tie the narrative together on the two diguanylate cyclases and the regulation of SPI-1. Without this, I found that reading about yedQ was a little distracting mid-way through the paper. The parts on the SicA(N70A) mutant could also benefit from some structure(s) to better understand how c-di-GMP regulates SicA, but this is probably beyond the scope of this work.

Response: We thank the reviewer for the insightful suggestions. Indeed, a *yedQ* mutant in the section on SicA, and the 3D structures of SicA and SicA-c-di-GMP complex to understand how c-di-GMP regulates SicA will further improve our work. Further work will be performed in the future to support the present work. In fact, the crystal structures of SicA and the SicA-c-di-GMP complex are now being resolved in the lab of Dr. Songying Ouyang (one of the corresponding authors in the present work).